# Group Fair Federated Learning via Stochastic Kernel Regularization

**Huzaifa Arif**                                                                 *arifh@rpi.edu*
*Rensselaer Polytechnic Institute, Troy, NY, United States*

**Keerthiram Murugesan**                                    *Keerthiram.Murugesan@ibm.com*
*IBM Research, Yorktown Heights, NY, United States*

**Pin-Yu Chen**                                                          *pin-yu.chen@ibm.com*
*IBM Research, Yorktown Heights, NY, United States*

**Alex Gittens**                                                                 *gittea@rpi.edu*
*Rensselaer Polytechnic Institute, Troy, NY, United States*

**Reviewed on OpenReview:** *https://openreview.net/forum?id=k8x44wVIs1*

## Abstract

Ensuring **group fairness** in federated learning (FL) presents unique challenges due to data heterogeneity and communication constraints. We propose Kernel Fair Federated Learning (`KFFL`), a novel algorithmic framework that incorporates group fairness into FL models using the Kernel Hilbert-Schmidt Independence Criterion (KHSIC) as a fairness regularizer. To address scalability, `KFFL` approximates the KHSIC with random feature maps, significantly reducing computational and communication overhead while achieving group fairness.

To address the resulting non-convex composite optimization problem, we propose `FedProxGrad`, a federated proximal gradient algorithm that guarantees convergence. Through experiments on standard benchmark datasets across both IID and Non-IID settings for regression and classification tasks, `KFFL` demonstrates its ability to balance accuracy and fairness effectively, outperforming existing methods by comprehensively exploring the accuracy–fairness trade-offs. Furthermore, we introduce `KFFL-TD`, a time-delayed variant that further reduces communication rounds, enhancing efficiency in decentralized environments. Code is available at `github.com/Huzaifa-Arif/KFFL`.

## 1 Introduction

Unintended unfairness in machine learning models poses significant challenges, particularly in decision-making processes that impact specific population groups (Dwork et al., 2012a; Agarwal et al., 2019b; Jalal et al., 2021). For instance, the COMPAS software, used in judicial decision-making for criminal offenses, has been shown to yield unjust outcomes disproportionately affecting the African American community (Dressel & Farid, 2018; Barenstein, 2019). Such findings underscore the need for model outputs to be fair with respect to protected demographic attributes like gender and race. Ensuring demographic fairness has therefore emerged as a critical challenge in machine learning, driving efforts to develop robust solutions for mitigating bias and ensuring equitable model deployment.

The literature on fair federated learning often uses the term **fairness** ambiguously to refer to either **client fairness** or **group fairness**. However, these concepts address distinct objectives. Works such as Chaudhury et al. (2022); Li et al. (2019); Donahue & Kleinberg (2021); Cui et al. (2021); Du et al. (2021) focus on client fairness in federated learning, aiming to ensure that the model performs equitably across clients' data (Mohri et al., 2019), thereby mitigating disparities arising from data heterogeneity among clients. In contrast, **group**

**fairness** (Ezzeldin et al., 2023; Papadaki et al., 2022) seeks to achieve fairness across different demographic groups. This involves establishing performance guarantees to ensure the global model is fair with respect to sensitive attributes, such as race or gender. More discussion on the related works is available in the Appendix A.1. This work addresses group fairness in the trained global model.

Recent advances in bias mitigation algorithms (Jalal et al., 2021; Correa et al., 2021; Agarwal et al., 2019b; Memarrast et al., 2023) largely depend on fairness regularizers incorporated into the training objective, which typically requires centralized access to data. However, in federated learning (FL) (Li et al., 2020), where data is distributed across clients, privacy regulations and bandwidth constraints often prohibit raw data sharing, making centralized approaches to group fairness impractical. Furthermore, implementing fairness regularizers in the FL setting presents additional challenges, including communication overhead, computational costs, and data heterogeneity, all of which complicate training a globally fair model.

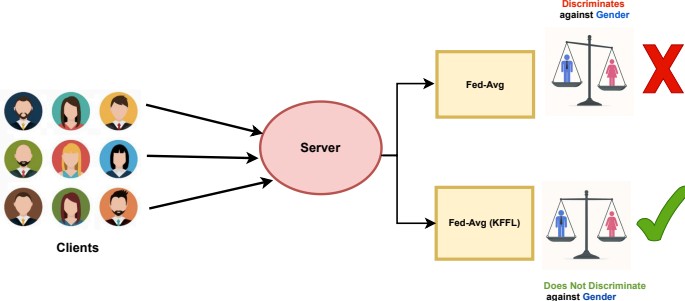

Figure 1: FEDAVG can result in models whose predictions are biased with respect to sensitive attributes such as race or gender. KFFL is a principled approach designed specifically to mitigate demographic bias when training a model in a distributed setting.

Consequently, prior efforts to achieve *group fairness* in federated settings have primarily focused on aligning local and global fairness metrics (Ezzeldin et al., 2023; Papadaki et al., 2022), often avoiding the direct incorporation of regularizer terms to ensure statistical group fairness. However, this approach faces challenges, as applying local debiasing mechanisms at individual clients alone is inadequate to ensure group fairness in the globally trained model.

We propose a novel approach to fair federated learning that integrates the Kernel Hilbert-Schmidt Independence Criterion (KHSIC) (Gretton et al., 2005b) as a fairness regularizer into FL. KHSIC is a powerful measure of statistical dependence capable of capturing complex, non-linear relationships between variables, making it well-suited for enforcing group fairness in a statistically principled manner.

The Hilbert-Schmidt Independence Criterion (HSIC) is an effective fairness regularizer in centralized regression models, as demonstrated in Pérez-Suay et al. (2017). KHSIC, the kernelized version of HSIC, captures complex, nonlinear dependencies among random variables through the use of kernel functions. Importantly, KHSIC provably quantifies statistical parity: a low KHSIC value between model outputs and sensitive attributes ensures approximate statistical parity (Kim & Gittens, 2021). Compared to the closely related Rényi correlation (Baharlouei et al., 2019b), KHSIC is more practical for measuring dependence because its empirical computation reduces to BLAS Level 3 matrix calculations. This makes KHSIC not only theoretically robust but also computationally efficient in practical applications.

However, directly applying KHSIC in a federated learning (FL) setup presents challenges, as it is computationally expensive and communication intensive due to the necessity of computing and exchanging large kernel matrices. To address these limitations, we leverage Random Feature Maps (RFMs) (Rahimi & Recht, 2007) to approximate kernel functions. This approximation significantly reduces both computational and communication costs, making the integration of KHSIC as a fairness regularizer more feasible and efficient in the FL setting.

Furthermore, to efficiently solve the distributed optimization problem incorporating the fairness regularizer, we introduce FEDPROXGRAD, a federated proximal gradient algorithm. This method ensures convergence for the non-convex composite optimization problems that arise from the integration of the fairness term.

Our contributions are as follows:

- We propose KFFL, a novel federated learning algorithm that incorporates group fairness using KHSIC as a fairness regularizer. To the best of our knowledge, this is the first work to adapt KHSIC for use in federated learning, addressing the unique challenges of the federated setting.

- To reduce the computational and communication overhead associated with KHSIC, we develop a communication-efficient approximation to the KHSIC using Random Feature Maps. This allows us to avoid transmitting large kernel matrices, reducing communication costs by orders of magnitude.

- We introduce FEDPROXGRAD, a federated proximal gradient algorithm that provides convergence guarantees for non-convex composite optimization problems. FEDPROXGRAD allows both terms of the composite objective to be non-convex, in contrast to prior works on federated composite optimization (Wang & Li, 2023; Bao et al., 2022; Yuan et al., 2021; Tran Dinh et al., 2021).

- We analyze the communication overhead of the resulting KHSIC-regularized fair federated learning algorithm, KFFL, and introduce a time-delayed variant, KFFL-TD, which further reduces communication rounds while maintaining performance. This makes the method more attractive in real-world FL applications where communication must be minimized.

- We conduct extensive experiments on standard benchmark datasets under both IID and Non-IID data distributions on both classification and regression tasks. Our results demonstrate that KFFL effectively balances the trade-off between accuracy and fairness, outperforming existing baselines and exploring the accuracy–fairness trade-offs more comprehensively. **We recommend that practitioners use binary search to choose the fairness hyperparameter $\lambda$ in KFFL and KFFL-TD to achieve their desired accuracy–fairness trade-off.**

## 2    Preliminaries

The goal of fair learning is to ensure that the model's output exhibits no undesirable dependencies on sensitive attributes.

We assume the observations are sampled i.i.d. from a joint distribution $\mathbb{P}(X, S, Y)$ to obtain training data $\{\mathbf{x}_i, \mathbf{s}_i, y_i\}_{i=1}^n$. Here, $\mathbf{x}_i$ contains the non-sensitive covariates, $\mathbf{s}_i$ contains sensitive covariates (which may be a binary scalar $s_i$ or multi-dimensional vector $\mathbf{s}_i$), and $y_i$ is the ground truth label for the $i$-th sample. This dataset is employed to train a classifier $f(\mathbf{x}; \boldsymbol{\omega})$, where $\boldsymbol{\omega}$ denotes the model parameters.[1]

As an example, consider the task of training a binary classifier for making hiring decisions. Here $\mathbf{x}$ consists of features that are ethically and legally allowable for use in making hiring decisions, $s$ represents *binary* sensitive features such as the individual's sex or marital status, and the ground truth decisions are $y_i \in \{0, 1\}$. The classifier makes predictions $\hat{y}_i = f(\mathbf{x}_i; \boldsymbol{\omega}) \in \{0, 1\}$, where 1 signifies a decision to hire.

### 2.1    Metrics for Group Fairness

We employ two canonical group–fairness notions—*statistical parity* and *equality of opportunity*—both in their classic **binary** form and in a more general **multi-group** form needed for intersectional or many–valued sensitive attributes.

---

[1]We follow the notation convention where boldface lowercase letters (e.g., $\mathbf{v}$) denote vectors, non-bold lowercase letters (e.g., $v$) represent scalars, and boldface uppercase letters (e.g., $\mathbf{M}$) signify matrices. For example, in the equation $\mathbf{v} = \mathbf{M}\mathbf{u} + b$, $\mathbf{v}$ and $\mathbf{u}$ are vectors, $\mathbf{M}$ is a matrix, and $b$ is a scalar.

### 2.1.1 Binary Classification Metrics

- **Statistical-Parity Difference (SPD).** A classifier attains statistical parity when the prediction is independent of the sensitive attribute, $\hat{y} \perp\!\!\!\perp s$ (Dwork et al., 2012b). For a *binary* attribute $s \in \{0, 1\}$ we measure the gap

$$\text{SPD} \;=\; \Pr(\hat{y} = 1 \mid s = 1) \;-\; \Pr(\hat{y} = 1 \mid s = 0). \tag{1}$$

- **Equality of Opportunity (EO).** Equality of opportunity requires identical true-positive rates across sensitive groups (Hardt et al., 2016); we similarly quantify EO by measuring the gap

$$\text{EO} \;=\; \Pr(\hat{y} = 1 \mid y = 1, \, s = 1) \;-\; \Pr(\hat{y} = 1 \mid y = 1, \, s = 0). \tag{2}$$

### 2.1.2 Multi-group Extension

When the sensitive attribute admits *more than two* categories or several attributes are combined (e.g. race *and* sex), we enforce the fairness constraints across *all* resulting groups. Let $\mathcal{S} = \{1, 2, \ldots, K\}$ with $K \geq 3$.

**Definition 1** (Multi-group Statistical Parity (Dwork et al., 2012b))**.** Define $r_s := \Pr(\hat{y} = 1 \mid s)$. The multi-group SPD gap is

$$\text{SPD}_{\text{multi}} \;=\; \max_{s \in \mathcal{S}} r_s \;-\; \min_{s \in \mathcal{S}} r_s \;=\; \max_{s, s' \in \mathcal{S}} |r_s - r_{s'}|. \tag{3}$$

**Definition 2** (Multi-group Equality of Opportunity (Hardt et al., 2016))**.** Let $\text{TPR}_s^{(y)} := \Pr(\hat{y} = 1 \mid y = 1, s)$.

$$\text{EO}_{\text{multi}} \;=\; \max_{s, s' \in \mathcal{S}} \left| \text{TPR}_s^{(y)} - \text{TPR}_{s'}^{(y)} \right|. \tag{4}$$

Lower values of $\text{SPD}_{\text{multi}}$ and $\text{EO}_{\text{multi}}$ indicate greater fairness; both reduce to Eqs. (1)–(2) when $K = 2$.

If several sensitive attributes are recorded (e.g. race *and* sex) one may: (i) concatenate them into an *intersectional* label $S = (\text{race}, \text{sex})$ and apply Definitions 1–2 to the resulting $K = \prod_j K_j$ groups, *or* (ii) compute the same gaps separately for each attribute and treat the model as fair only if fairness with respect to *each* attribute meets the chosen tolerance. The choice between joint and per-attribute constraints depends on the policy requirements of the application.

Although our algorithm is designed to target SPD, as is common in the literature we use *both* SPD and EO to evaluate classifier fairness empirically in Section 6. We report both the binary metrics (Eqs. (1)–(2)) and their multi-group counterparts (Eqs. (3)–(4)).

### 2.1.3 Fairness Metrics for Regression

Fair regression has been extensively studied in centralized settings (Chzhen et al., 2020a; Agarwal et al., 2019a; Chzhen et al., 2020b). However, the training of fair regression models in the distributed settings remains underexplored. Fairness metrics like EO and SPD that are designed for classifiers are not applicable in this context. Instead, the **Kolmogorov-Smirnov (KS)** distance has been employed to evaluate regression fairness (Chzhen et al., 2020a). This metric captures the maximum disparity between the distributions of the model's predictions for different sensitive groups.

To formally define the KS distance, we denote the set of indices of samples belonging to a sensitive group $s \in \mathcal{S}$ by

$$\mathcal{I}^s = \{i \in \{1, 2, \ldots, n\} : s_i = s\}.$$

The empirical cumulative distribution function (CDF) of the model's predictions for group $s$ is

$$F^s(t; \boldsymbol{\omega}) = \frac{1}{|\mathcal{I}^s|} \sum_{i \in \mathcal{I}^s} \mathbb{1}\{f(\mathbf{x}_i; \boldsymbol{\omega}) \le t\}, \tag{5}$$

where $f(\mathbf{x}_i; \boldsymbol{\omega})$ is the model's prediction for input $\mathbf{x}_i$.

The KS distance between the predictions for two distinct sensitive groups $s$ and $s'$ is

$$\text{KS}(\boldsymbol{\omega}) = \max_{s,s' \in \mathcal{S}} \sup_{t \in \mathbb{R}} \left| F^s(t; \boldsymbol{\omega}) - F^{s'}(t; \boldsymbol{\omega}) \right|. \tag{6}$$

A smaller KS distance indicates lower disparity between the regression results across the different sensitive groups. In fact, the KS distance is commonly used in the KS test, a statistical test for the equality of one-dimensional probability distributions.

Appendix F provides experimental results on using the KHSIC regularizer for regularization tasks, where the resulting fairness is quantified using the KS distance.

## 2.2 The Kernel Hilbert-Schmidt Independence Criterion (KHSIC)

The **Kernel Hilbert-Schmidt Independence Criterion** (**KHSIC**) of Gretton et al. (2005a) is key to our approach to ensuring statistical parity. The KHSIC is predicated on the observation that $\hat{y}$ and $s$ are independent if and only if *every* function of $\hat{y}$ is uncorrelated with *every* function of $s$.

Given two Reproducing Kernel Hilbert Spaces $\mathcal{F}$ and $\mathcal{G}$, the (population) KHSIC is the Hilbert-Schmidt norm of the centered cross-covariance operator $\mathcal{C}_{\hat{y}s}$ of $\hat{y}$ and $s$ (Gretton et al., 2005a):

$$\psi_{\text{pop}}(\hat{y}, s) = \|\mathcal{C}_{\hat{y}s}\|_{\text{HS}}^2 = \sum_i \sum_j \text{Cov}(f_i(\hat{y}), g_j(s))^2. \tag{7}$$

Here $\{f_i\}_{i=1}^{\infty}$ and $\{g_j\}_{j=1}^{\infty}$ are orthonormal bases for $\mathcal{F}$ and $\mathcal{G}$, respectively. It follows from the bilinearity of the covariance operator that the KHSIC quantifies the dependence of $\hat{y}$ and $s$ by bounding the covariance of every function of $\hat{y}$ in $\mathcal{F}$ with every function of $s$ in $\mathcal{G}$.

Under some regularity conditions on $\mathcal{F}$ and $\mathcal{G}$ (universality), the population KHSIC is zero if and only if $\hat{y}$ and $s$ are independent. Thus when the KHSIC is zero,

$$\text{SPD} \;=\; \Pr(\hat{y} = 1 \mid s = 1) \;-\; \Pr(\hat{y} = 1 \mid s = 0) = \Pr(\hat{y} = 1) - \Pr(\hat{y} = 1) = 0,$$

so the classifier $\hat{y}$ achieves statistical parity. More generally, the population KHSIC gives an upper bound on the total variation distance between the joint distribution $\mathbb{P}_{\hat{y},s}$ and the product of the marginals $\mathbb{P}_{\hat{y}} \otimes \mathbb{P}_s$, and thus quantifies the dependence of $\hat{y}$ and $s$ (Kim & Gittens, 2021).

The KHSIC is more practically useful as a measure of dependence than the closely related Rényi correlation: the latter is defined in terms of the probability density functions (PDFs) of $\hat{y}$ and $s$. This approach does not scale to high-dimensional inputs because accurately estimating the PDFs of $\hat{y}$ and $s$ requires an exponential number of observations relative to their dimensionality. By comparison, empirical estimation of the KHSIC reduces to simple and scalable linear algebraic computations. The empirical KHSIC is derived in Gretton et al. (2005a):

$$\psi_{\text{emp}}(\hat{\mathbf{y}}, \mathbf{s}) = \frac{1}{(n-1)^2} \text{Tr} \left( \mathbf{H} \mathbf{K_s} \mathbf{H}^2 \mathbf{K_{\hat{y}}} \mathbf{H} \right), \tag{8}$$

where

$$\mathbf{K_{\hat{y}}} = [\kappa_{\mathcal{F}}(\hat{y}_i, \hat{y}_j)]_{i,j=1}^n, \quad \mathbf{K_s} = [\kappa_{\mathcal{G}}(\mathbf{s}_i, \mathbf{s}_j)]_{i,j=1}^n, \quad \mathbf{H} = \mathbf{I} - \frac{1}{n} \mathbf{1} \mathbf{1}^{\top}.$$

The functions $\kappa_{\mathcal{F}}$ and $\kappa_{\mathcal{G}}$ are the kernels associated with the RKHSes $\mathcal{F}$ and $\mathcal{G}$, respectively, and the matrices $\mathbf{K_{\hat{y}}}$ and $\mathbf{K_s}$ are $n \times n$ kernel matrices evaluated on the predictions and the corresponding sensitive attributes. [2] Gretton et al. (2005a) show that $\psi_{\text{emp}}(\hat{\mathbf{y}}, \mathbf{s})$ is a $\frac{1}{\sqrt{n}}$-consistent estimator of $\psi_{\text{pop}}(\hat{y}, s)$.

We propose to measure the fairness (in the sense of statistical parity) of models by using the empirical KHSIC:

$$\psi(\boldsymbol{\omega}; \mathbf{X}, \mathbf{S}) = \frac{1}{(n-1)^2} \text{Tr} \left( \mathbf{H} \mathbf{K_s} \mathbf{H} \mathbf{K_{\hat{y}}} \mathbf{H} \right). \tag{9}$$

---

[2]For notational brevity, we use the notation $\kappa$ to refer to two potentially different kernel functions on the features and the sensitive variables.

The notation $\psi(\boldsymbol{\omega}; \mathbf{X}, \mathbf{S})$ emphasizes that the fairness measure depends on $\boldsymbol{\omega}$ through the predictions $\hat{\mathbf{y}} = f(\mathbf{X}; \boldsymbol{\omega})$.

### 2.3 The Fair Learning Objective

To learn a fair model $\boldsymbol{\omega}_\star$ by using $\psi(\boldsymbol{\omega}; \mathbf{X}, \mathbf{S})$ as a regularizer, we solve the optimization problem

$$\boldsymbol{\omega}_\star = \text{argmin}_{\boldsymbol{\omega}} \frac{1}{n} \sum_{i=1}^{n} \ell(y_i, f(\mathbf{x}_i; \boldsymbol{\omega})) + \lambda \psi(\boldsymbol{\omega}; \mathbf{X}, \mathbf{S}). \tag{10}$$

The first term in the objective function represents the loss $\ell(y_i, f(\mathbf{x}_i; \boldsymbol{\omega}))$, which quantifies the discrepancy between the model's predictions and the ground truth labels. This loss function may be instantiated as the cross-entropy loss for classification tasks or the mean squared error (MSE) for regression tasks. The second term acts as a fairness regularizer by measuring the dependence between the model's predictions and the sensitive attributes using the KHSIC criterion. The fairness parameter $\lambda$ controls the trade-off between optimizing predictive performance and enforcing fairness.

Solving (10) in a centralized setting is conceptually straightforward but computationally challenging because, at each iteration, it involves computing forward and backward passes (for backpropagation) through matrices $\mathbf{K}_{\hat{\mathbf{y}}}(\boldsymbol{\omega})$ of size $n \times n$. The computational burden is compounded in the federated learning setting by the additional communication burden: clients need to communicate these $n \times n$ matrices to compute their local contributions to the gradients. Hence, a straightforward adaptation of this centralized approach to a distributed setting is prohibitive from the perspectives of both computation and communication. We address this challenge in Section 4.

In addition to the challenges posed by the specific form of $\psi$, the composite structure of optimization problem (10) is itself a challenge, as both terms in the objective are non-convex. We introduce `FedProxGrad` in the next section (Section 3), to address this challenge.

## 3 Composite Optimization in the Federated Setting

Employing the fair ML formulation of equation (10) in a federated setting requires solving a federated composite optimization (FCO) problem that can be written in the standard form

$$\text{argmin}_{\boldsymbol{\omega}} \sum_{i=1}^{m} \ell^i(\boldsymbol{\omega}) + \psi(\boldsymbol{\omega}),$$

where the $\ell^i$ are data-fitting terms local to each client, and $\psi$ is the global fairness regularizer. **For brevity, we have absorbed the regularization constant $\lambda$ into $\psi$.** We denote the sum of local-data fitting terms with $\ell(\boldsymbol{\omega}) = \sum_{i=1}^{m} \ell^i(\boldsymbol{\omega})$ and the composite objective with $F(\boldsymbol{\omega}) = \ell(\boldsymbol{\omega}) + \psi(\boldsymbol{\omega})$. In Equation (10) the fairness term $\psi$ is non-convex and the data fitting term, $\ell$, may also be non-convex. This section provides an algorithm for solving the resulting non-convex FCO problem.

Several existing works provide algorithms for the FCO problem–Wang & Li (2023); Bao et al. (2022); Yuan et al. (2021) develop algorithms that require the $\ell^i$ and $\psi$ to be convex, while the algorithm of Tran Dinh et al. (2021) requires only $\psi$ to be convex—but, to our knowledge, no extant optimization algorithms for FCO guarantees convergence for problems where the $\ell^i$ and $\psi$ are both non-convex. To fill this gap, we introduce `FedProxGrad`, a federated proximal gradient descent algorithm. Unlike prior FCO algorithms, `FedProxGrad` imposes no convexity requirements.

The `FedProxGrad` algorithm, described in Algorithm 1, extends the stochastic proximal gradient algorithm in a straightforward manner from the centralized setting to the federated setting. We note that the `FedProxGrad` algorithm differs significantly from the `FedProx` algorithm: `FedProx` is a *federated proximal point algorithm* designed to minimize a *single objective*, while `FedProxGrad` is a *federated proximal gradient algorithm* designed to minimize a *composite objective*.

---

**Algorithm 1** Federated Proximal Gradient Descent (`FedProxGrad`)

---

1: **Input:** $\boldsymbol{\omega}_0$, $T$, $\alpha$
2: **for** $t = 0, \cdots, T-1$ **do**
3:     Server computes $\mathbf{g}_t$, a stochastic gradient estimator for $\nabla\psi(\boldsymbol{\omega}_t)$, and computes $\boldsymbol{\omega}_{t+1/2} = \boldsymbol{\omega}_t - \alpha\mathbf{g}_t$
4:     Server sends $\boldsymbol{\omega}_{t+1/2}$ to clients
5:     Each client for $i \in \{1, \ldots, m\}$ computes

$$\boldsymbol{\omega}_{t+1}^i = \operatorname{argmin}_{\boldsymbol{\omega}} \ell^i(\boldsymbol{\omega}) + \frac{1}{2\alpha}\|\boldsymbol{\omega} - \boldsymbol{\omega}_{t+1/2}\|_2^2$$

6:     Each client returns $\boldsymbol{\omega}_{t+1}^i$ back to the server
7:     Server aggregates the device models to form

$$\boldsymbol{\omega}_{t+1} = \frac{1}{m}\sum_{i=1}^{m}\boldsymbol{\omega}_{t+1}^i.$$

8: **end for**

---

Nonetheless, the analysis of the convergence of `FedProxGrad` follows closely that of the convergence rate of `FedProx`, so we introduce the same notions used in its convergence (Li et al., 2020).

**Definition 3** ($\gamma$-suboptimality). Let $\ell_t^i(\boldsymbol{\omega}) = \ell^i(\boldsymbol{\omega}) + \frac{1}{2\alpha}\|\boldsymbol{\omega} - \boldsymbol{\omega}_{t+1/2}\|^2$ (see Algorithm 1 for the definition of $\boldsymbol{\omega}_{t+1/2}$). Given $\gamma \in [0, 1]$, a point $\widehat{\boldsymbol{\omega}}$ is a $\gamma$-suboptimal solution of $\operatorname{argmin}_{\boldsymbol{\omega}}\ell_t^i(\boldsymbol{\omega})$ if $\|\nabla\ell_t^i(\widehat{\boldsymbol{\omega}})\| \leq \gamma\|\nabla\ell_t^i(\boldsymbol{\omega}_t)\|$. Smaller $\gamma$ correspond to higher accuracy.

This definition of $\gamma$-suboptimality slightly differs from that used in Li et al. (2020), to facilitate the analysis of a composite objective. This condition on the local solvers ensures that the local solution for the $(t+1)$th iterate is a factor of $\gamma$ less suboptimal than the $t$th iterate. This condition is agnostic to the particular solvers employed, and can be achieved using deterministic or randomized solvers that use full gradients or stochastic gradients.

**Definition 4** ($(G, B)$-Bounded Dissimilarity (Definition A1 of Karimireddy et al. (2020))). The local functions $\ell^i$ are $(G, B)$-boundedly dissimilar at $\boldsymbol{\omega}$ if $\mathbb{E}_i\|\nabla\ell^i(\boldsymbol{\omega}) + \nabla\psi(\boldsymbol{\omega})\|^2 \leq B^2\|\nabla F(w)\|^2 + G^2$.

This condition is standard Li et al. (2020); Karimireddy et al. (2020), and ensures that local progress on the individual clients can be translated to global progress. The bounded dissimilarity condition mirrors common optimization assumptions (e.g., bounded Lipschitz or bounded Hessian assumptions): when it is violated by taking $B \to \infty$, the allowed step size shrinks, and the convergence rate degrades[3].

**Theorem 1.** *Assume that the functions $\ell^i$, $\psi$, and $F$ are $L$-smooth; the functions $\ell^i$ are $L_-$-weakly convex; $F$ is bounded below by a constant $F^\star$; the bounded dissimilarity condition (Definition 4) holds with $G = 0$; and that the stochastic gradient estimate for the fairness regularizer satisfies $\mathbb{E}[\mathbf{g}_t \,|\, \boldsymbol{\omega}_t] = \nabla\psi(\boldsymbol{\omega}_t)$ and*

$$\mathbb{E}[\|\mathbf{g}_t - \nabla\psi(\boldsymbol{\omega})\|_2^2 \mid \boldsymbol{\omega}] \leq \sigma^2$$

*for all $\boldsymbol{\omega}$. If the local solvers on each client ensure $\gamma$ suboptimal solutions (Definition 3) with parameter $\gamma \leq \frac{1}{8(B+1)}$ and the global stepsize is chosen to satisfy*

$$\alpha < \min\left\{\frac{1}{20}, \frac{1}{2L_-}, \frac{1}{120L(B+1)}, \frac{1}{5LB^2}\right\},$$

*then the sequence of iterates generated by FedProxGrad satisfies*

$$\frac{1}{T}\sum_{t=0}^{T-1}\mathbb{E}\left[\|\nabla F(\boldsymbol{\omega}_t)\|_2^2\right] \leq \frac{F(\boldsymbol{\omega}_0) - F^\star}{\alpha T} + 4\sigma^2.$$

---

[3]A reviewer astutely pointed out that the bounded dissimilarity assumption may possibly be avoided at the cost of a reduced convergence rate by using the approach of (Yuan & Li, 2022).

This result shows that, for an appropriate choice of hyperparameters, the `FedProxGrad` algorithm converges at a rate of $\frac{1}{T}$ up to the noise level of the stochastic gradient. A proof is provided in the Appendix A.2. We note that only recently has a nonasymptotic rate of convergence of stochastic proximal gradient algorithms for fully nonconvex composite optimization problems been established in the shared memory setting (Xu et al., 2019). Theorem 1 shows that a natural generalization of the stochastic proximal gradient algorithm to the fully nonconvex federating setting also converges at a sublinear rate.

## 4 Communication-Efficient Kernel Regularized Fair Learning

In the centralized setting, kernel methods pose a computational challenge due to their inherent complexity, requiring optimization with a kernel matrix incurring a computational cost of up to $\mathcal{O}(n^3)$. One line of research for reducing this burden, starting with the seminal work of Rahimi & Recht (2007), uses random feature maps (RFMs). A random feature map for a shift-invariant kernel function $\kappa$ is a random function $\phi : \mathbb{R}^p \to \mathbb{R}^D$ that is constructed to satisfy $\kappa(\mathbf{x}, \mathbf{y}) = \mathbb{E}\langle \phi(\mathbf{x}), \phi(\mathbf{y}) \rangle$ for any two vectors $\mathbf{x}$ and $\mathbf{y}$, where the expectation is taken over the randomness in $\phi$.

RFMs enable the efficient formation of randomized low-rank approximations to kernel matrices. In particular, if the rows of $\mathbf{Z}_S \in \mathbb{R}^{n \times D}$ consist of the application of an RFM $\phi$ to the $\mathbf{s}_i$, and the rows of $\mathbf{Z}_{f(\boldsymbol{\omega})} \in \mathbb{R}^{n \times D}$ likewise consist of the application an RFM to the observed $\mathbf{x}_i$, then

$$\mathbf{K}_S = \mathbb{E}[\mathbf{Z}_S \mathbf{Z}_S^T] \quad \text{and} \quad \mathbf{K}_{f(\boldsymbol{\omega})} = \mathbb{E}[\mathbf{Z}_{f(\boldsymbol{\omega})} \mathbf{Z}_{f(\boldsymbol{\omega})}^T], \tag{11}$$

and the variance of the approximations go down as the number of random features $D$ increases, so $\mathbf{Z}_S \mathbf{Z}_S^T$ and $\mathbf{Z}_{f(\boldsymbol{\omega})} \mathbf{Z}_{f(\boldsymbol{\omega})}^T$ are principled randomized low-rank approximations to the corresponding kernel matrices. A substantial body of work has demonstrated that these approximations exhibit theoretically and empirically similar performance to full kernel matrices (Hamid et al., 2014; Rahimi & Recht, 2007; Yu et al., 2016). More details on the construction of randomized feature maps is given in Appendix G

We utilize these randomized low-rank approximations to efficiently compute principled approximations to the regularizer $\psi(\boldsymbol{\omega})$ in Equation 10. Note that $\mathbf{Z}_{f(\boldsymbol{\omega})}$ and $\mathbf{Z}_S$ have dimensions $n \times D$ where $D \ll n$. Specifically, we utilize *the Orthogonal Random Feature Maps (ORFM)* of (Yu et al., 2016).

The first crucial observation is that by using RFMs, one need only communicate a $D \times D$ matrix to approximate $\psi(\boldsymbol{\omega})$ and its gradient $\mathbf{g}(\boldsymbol{\omega})$, rather than communicating two $n \times n$ kernel matrices.

**Theorem 2.** *Let $\mathbf{Z}_{f(\boldsymbol{\omega})}$ and $\mathbf{Z}_S$ be the $n \times D$ matrices constructed using RFMs, corresponding respectively to the prediction kernel $\mathbf{K}_{f(\boldsymbol{\omega})}$ and the sensitive attribute kernel $\mathbf{K}_S$. Then $\psi(\boldsymbol{\omega}) = \mathbb{E}\left[\|\mathbf{G}(\boldsymbol{\omega})\|_F^2\right]$ where $\mathbf{G}(\boldsymbol{\omega}) = \mathbf{Z}_S^\top \mathbf{Z}_{f(\boldsymbol{\omega})} - n \boldsymbol{\mu}_s \boldsymbol{\mu}_f^\top(\boldsymbol{\omega}) \in \mathbb{R}^{D \times D}$. Here, $\boldsymbol{\mu}_s$ is the mean over the rows of $\mathbf{Z}_S$ and $\boldsymbol{\mu}_f$ is the mean over the rows of $\mathbf{Z}_{f(\boldsymbol{\omega})}$.*

**Corollary 1.** $\mathbf{g}(\boldsymbol{\omega}) = \nabla\|\mathbf{G}(\boldsymbol{\omega})\|_F^2$ *is an unbiased stochastic estimate of $\nabla\psi(\boldsymbol{\omega})$.*

Proofs of these results, consisting of basic linear algebraic manipulations, are supplied in Appendices A.3 and A.5, respectively. In the remainder of this section, we detail how $\mathbf{G}(\boldsymbol{\omega})$, and consequently the gradient estimate $\mathbf{g}(\boldsymbol{\omega})$, can be computed efficiently in the federated setting.

The next important observation is that in the federated setting, the RFMs $\mathbf{Z}_{f(\boldsymbol{\omega})}$ and $\mathbf{Z}_S$ are naturally partitioned across the clientss:

$$\mathbf{Z}_{f(X)} = \begin{pmatrix} \mathbf{Z}_{f(X),1} \\ \vdots \\ \mathbf{Z}_{f(X),m} \end{pmatrix} \quad \text{and} \quad \mathbf{Z}_S = \begin{pmatrix} \mathbf{Z}_{S,1} \\ \vdots \\ \mathbf{Z}_{S,m} \end{pmatrix}, \tag{12}$$

where $\mathbf{Z}_{f(X),i}, \mathbf{Z}_{S,i} \in \mathbb{R}^{n_i \times D}$. Here $n_i$ is the number of data points on client $i$, so $n = \sum_{i=1}^m n_i$. This observation allows each client to, independently from the other clients, efficiently compute its contribution to the feature interaction matrix $\mathbf{G}(\boldsymbol{\omega})$.

**Lemma 1.** *The global feature interaction matrix* $\mathbf{G}(\boldsymbol{\omega})$ *can be partitioned into local interactions as*

$$\mathbf{G}(\boldsymbol{\omega}) = \sum_{i=1}^{m} \mathbf{Z}_{S,i}^{\top} \mathbf{Z}_{f(X),i} - n \left( \frac{1}{n} \sum_{i=1}^{m} n_i \boldsymbol{\mu}_{S,i} \right) \left( \frac{1}{n} \sum_{i=1}^{m} n_i \boldsymbol{\mu}_{f,i} \right)^{\top},$$

*where* $\boldsymbol{\mu}_{S,i}$ *and* $\boldsymbol{\mu}_{f,i}$ *are the row averages of* $\mathbf{Z}_{S,i}$ *and* $\mathbf{Z}_{f(X),i}$, *respectively.*

As a consequence of this result, the server can compute the global interaction term $\mathbf{G}(\boldsymbol{\omega})$ after each client transmits its local feature interaction matrix $\mathbf{Z}_{S,i}^{\top} \mathbf{Z}_{f(X),i} \in \mathbb{R}^{D \times D}$, along with its local average feature vectors $\boldsymbol{\mu}_{S,i}, \boldsymbol{\mu}_{f,i} \in \mathbb{R}^{D}$ to the server. This approach facilitates the computation of $\mathbf{G}(\boldsymbol{\omega})$ and $\mathbf{g}(\boldsymbol{\omega})$ in a manner that does not require communication between the clients.

**Lemma 2.** *The unbiased estimate of the gradient of the fairness regularizer* $\psi$ *from Corollary 1 can be partitioned into local interactions:*

$$\mathbf{g}(\boldsymbol{\omega}) = \sum_{i=1}^{m} \mathbf{J}_{\boldsymbol{\Omega}_i}(\boldsymbol{\omega})^{T} \mathbf{G}(\boldsymbol{\omega}),$$

*where* $\boldsymbol{\Omega}_i(\boldsymbol{\omega}) = \mathbf{Z}_{S,i}^{\top} \mathbf{Z}_{f(X),i}(\boldsymbol{\omega}) - n_i \boldsymbol{\mu}_{S,i} \boldsymbol{\mu}_{f,i}^{\top}$ *and* $\mathbf{J}_{\boldsymbol{\Omega}_i}(\boldsymbol{\omega})$ *is the Jacobian of* $\boldsymbol{\Omega}_i$ *with respect to the model parameters* $\boldsymbol{\omega}$.

This result states that once the server computes the global interaction matrix $\mathbf{G}(\boldsymbol{\omega})$ and distributes it to the clients, each client can then independently compute its contribution to a stochastic estimate of the gradient of the fairness regularizer $\psi(\boldsymbol{\omega})$. Proofs of these lemmata are provided in Appendices A.4 and A.6.

Therefore, **by using random feature maps, instead of transmitting** $n \times n$ **matrices, only** $D \times D$ **matrices need be sent to compute unbiased approximations to** $\psi$ **and** $\nabla \psi$. This results in a substantial reduction in communication costs. For instance, consider training a fair model on the ADULT dataset ($n = 32K$). Choosing $D = 1024$ (the dimensionality of the RFMs used in our experimental evaluations), computing $\psi(\boldsymbol{\omega})$ exactly requires communicating $32K \times 32K$ matrices, compared to $1K \times 1K$ matrices when RFMs are employed to estimate $\psi(\boldsymbol{\omega})$. In this example, the communication cost is reduced by three orders of magnitude!

The use of randomized projections to reduce the cost of communication is well-established. We note in particular (Han et al., 2024): this work uses randomized projections to help in their aim of achieving performance fairness (see Definition 1 of (Han et al., 2024)), by minimizing the variance in the client objectives; this concept of fairness differs from that of group fairness, which is our focus. Our contribution is in applying random projections to achieve group fair federated learning in a statistically principled manner.

## 5 The `KFFL` algorithm for `K`ernel-regularized `F`air `F`ederated `L`earning

We now introduce the `KFFL` algorithm, which uses the `FedProxGrad` method to implement kernel-regularized fair learning using the approximation to $\psi$ introduced in the previous section. A time-delayed variant that uses stale fairness gradient information to incur one less round of communication per iteration, `KFFL-TD`, is presented in Appendix C.

The `KFFL` algorithm is detailed in Algorithm 2, which gives the client-side procedure, and Algorithm 3, which gives the server-side process. Equations (13) through 16 are referenced in these algorithms. At a high level, `KFFL` uses federated composite optimization to fit a fair model in three rounds:

**FAIR1** At the start of the $(t+1)$-th iteration, the clients use the RFMs to compute the local terms $\Phi^i(\boldsymbol{\omega}_t)$ needed to compute the interaction matrix $\mathbf{G}(\boldsymbol{\omega}_t)$ for the current global model $\boldsymbol{\omega}_t$, and communicate them to the server. Clients also store the local interaction term for next round $\mathbf{M}^i(\boldsymbol{\omega}_t)$. The server combines the local terms to compute the interaction matrix $\mathbf{G}(\boldsymbol{\omega}_t)$ on the current global model, and returns these to the clients.

**FAIR2** The clients use $\mathbf{G}(\boldsymbol{\omega}_t), \boldsymbol{\mu}_s(t) \boldsymbol{\mu}_f(t)^{\top}$ from the server and the local interaction term $\mathbf{M}^i(\boldsymbol{\omega}_t)$ previously computed to compute their contribution to the stochastic estimate of the fairness gradient and communicate these to the server.

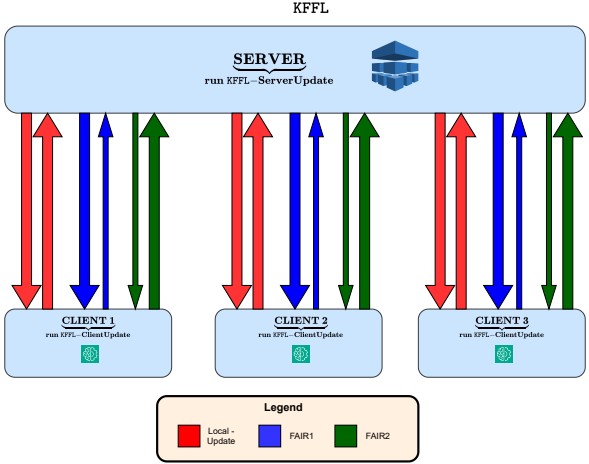

Figure 2: The communication pattern for KFFL. Different colors correspond to the **FAIR1**, **FAIR2** and **Local Update** sub-rounds of KFFL. The direction of the arrows indicate an uplink or downlink communication and the width of each arrowhead highlights the communication cost in each sub-round. Thicker lines indicate (large) communication costs on the order of the size of $\boldsymbol{\omega}$, while thinner lines represent communication costs on the order of $D^2$.

**Update** [4]The rest of the iteration implements FedProxGrad: the server computes the gradient estimate for $\psi$ and sends $\boldsymbol{\omega}_{t+\frac{1}{2}} = \boldsymbol{\omega}_t - \mathbf{g}_t$ to the clients. The clients locally update their models $\boldsymbol{\omega}^i_{t+1}$ and send them back to the server, which then computes the next global model $\boldsymbol{\omega}_{t+1}$.

Figure 7 graphically depicts the rounds of communication in each iteration of the KFFL algorithm. The following expressions are used in Algorithms 3 and 2.

Equations used in KFFL/KFFL-TD

$$\mathbf{G}(\boldsymbol{\omega}) = \mathbf{Z}_s^\top \mathbf{Z}_f(\boldsymbol{\omega}) - n\boldsymbol{\mu}_s \boldsymbol{\mu}_f^\top(\boldsymbol{\omega}) \tag{13}$$

$$\mathbf{M}^i(\boldsymbol{\omega}) = \mathbf{Z}_s^\top \mathbf{Z}_{f,i}(\boldsymbol{\omega}) \tag{14}$$

$$\boldsymbol{\Omega}_i(\boldsymbol{\omega}) = \mathbf{M}^i(\boldsymbol{\omega}) - n_i \boldsymbol{\mu}_s \boldsymbol{\mu}_f^\top \tag{15}$$

$$\mathbf{g}^i(\omega) = \mathbf{J}_{\boldsymbol{\Omega}_i}(\omega)^T \mathbf{G}(\boldsymbol{\omega}) \tag{16}$$

$$\boldsymbol{\mu}_s = \left( \frac{1}{n} \sum_{i=1}^m n_i \boldsymbol{\mu}_{s,i} \right) \tag{17}$$

$$\boldsymbol{\mu}_f = \left( \frac{1}{n} \sum_{i=1}^m n_i \boldsymbol{\mu}_{f,i} \right)^\top \tag{18}$$

$$\boldsymbol{\mu}_{s,i}^\top = \frac{1}{n_i} \mathbf{1}^\top \mathbf{Z}_{s,i} \tag{19}$$

$$\boldsymbol{\mu}_{f,i}^\top = \frac{1}{n_i} \mathbf{1}^\top \mathbf{Z}_{f,i} \tag{20}$$

$$\boldsymbol{\omega}_{t+1} = \operatorname{argmin}_{\boldsymbol{\omega}} \left[ f_i(\boldsymbol{\omega_t}) + \frac{1}{2\alpha_t} \|\boldsymbol{\omega} - \boldsymbol{\omega}_t\|_2^2 \right] \tag{21}$$

---

[4]Here **Update** is written as a short to the **Local Update** flag

---

**Algorithm 2** `KFFL` – Client Side

---

**Input**: $(\mathbf{ROUND}, ..)$

1: **if ROUND** = **FAIR1 then**
2:     Clients compute $\mathbf{M}^i(\boldsymbol{\omega}_t)$ (see Equation 14) using shared random seed $\zeta$ to generate their RFM, $\boldsymbol{\mu}_{f,i}(t)$ using Equation 20 and $\boldsymbol{\mu}_{s,i}(t)$ using Equation 19.
3:     Combine terms $\Phi^i(\boldsymbol{\omega}_t) = \{\mathbf{M}^i(\boldsymbol{\omega}_t), \boldsymbol{\mu}_{s,i}, \boldsymbol{\mu}_{f,i}\}$
4:     **Return**: $\Phi^i(\boldsymbol{\omega}_t)$
5: **else if ROUND** = **FAIR2 then**
6:     Client compute local interaction for gradients using Equation 15 to get $\boldsymbol{\Omega}_i(\boldsymbol{\omega}_t)$
7:     The clients then compute the local gradient $\mathbf{g}^i(\boldsymbol{\omega}_t)$ using Equation 16 from $\Lambda(\boldsymbol{\omega}_t)$
8:     **Return**: $\mathbf{g}^i(\boldsymbol{\omega}_t)$
9: **else if ROUND** = **Local Update then**
10:     Clients do a local update on $\boldsymbol{\omega}_{t+1/2}$ following Equation 21 to get $\boldsymbol{\omega}_{t+1}^i$
11:     **Return**: $\boldsymbol{\omega}_{t+1}^i$
12: **end if**

---

**Algorithm 3** `KFFL` – Server Side

---

1: $\boldsymbol{\omega} = \boldsymbol{\omega}_0$ {This is the initial model}
2: $t \leftarrow 0$
3: **while** $\boldsymbol{\omega}$ not converged **do**
4:     **for all** $i = 1, \ldots, m$ in parallel **do**
5:         Generation of random seed $\zeta$
6:         $\Phi^i(\boldsymbol{\omega}_t) = $ **Client Update**(**FAIR1**, $\boldsymbol{\omega}_t, \zeta$)
7:     **end for**
8:     $\Phi(\boldsymbol{\omega}_t) = \{\Phi^i(\boldsymbol{\omega}_t)\}_{i=1}^m$
9:     From $\Phi(\boldsymbol{\omega}_t)$ compute $\mathbf{G}(\boldsymbol{\omega}_t)$ using Equation 13 ; $\boldsymbol{\mu}_s(t)$ using Equation 17 and $\boldsymbol{\mu}_f(t)$ using Equation 18
10:     **for all** $i = 1, \ldots, m$ in parallel **do**
11:         $\Lambda(\boldsymbol{\omega}_t) = \{\mathbf{FAIR2}, \mathbf{G}(\boldsymbol{\omega}_t), \boldsymbol{\mu}_s(t)\boldsymbol{\mu}_f(t)^\top\}$
12:         $\mathbf{g}^i(\boldsymbol{\omega}_t) = $ **Client Update**($\Lambda(\boldsymbol{\omega}_t)$)
13:     **end for**
14:     $\boldsymbol{\omega}_{t+1/2} \leftarrow \boldsymbol{\omega}_t - \sum_{i=1}^m \mathbf{g}^i(\boldsymbol{\omega}_t)$
15:     **for all** $i = 1, \ldots, m$ in parallel **do**
16:         $\boldsymbol{\omega}_{t+1}^i = $ **Client Update**(**Local Update**, $\boldsymbol{\omega}_{t+1/2}$)
17:     **end for**
18:     $\boldsymbol{\omega}_{t+1} \leftarrow$ average $\left(\boldsymbol{\omega}_{t+1}^i\right)$
19: **end while**

---

The following section (6) evaluates the empirical performance of `KFFL` and `KFFL-TD` against baseline federated learning algorithms designed to mitigate demographic bias, utilizing the fairness metrics introduced in Section 2. Furthermore, we examine the communication costs of `KFFL` and `KFFL-TD` relative to these baselines.

## 6 Experimental Evaluation

In this section, we evaluate the performance of our methods, KFFL and KFFL-TD, at achieving group fairness in the federated setting. Fairness is assessed using statistical parity, (1), and equal opportunity, (2), for classification models and the KS distance, (6), for regression models. While most of the work in fair federated learning has explored fairness algorithms that aim to achieve *client fairness* (i.e., consistent performance across clients), as in Cui et al. (2021) and Du et al. (2021), such algorithms *do not directly target statistical group fairness*. Thus, these algorithms are not suitable benchmarks for our approach, as we focus explicitly on addressing statistical group fairness in the federated learning.

We compare KFFL against three baseline methods: `FedAvg` McMahan et al. (2017); the `MinMax` algorithm of Papadaki et al. (2022), which aims to optimize model performance for the worst-performing demographic group; the `FairFed` algorithm of Ezzeldin et al. (2023), where clients convey localized fairness metrics and the server optimizes weighting coefficients by minimizing the contribution of the poorest-performing client

for a chosen fairness metric. The framework of `FairFed` allows for different local bias mitigation algorithms. We use `FairFed` with the best performing local bias mitigation algorithm, `FairBatch` (Roh et al., 2020). The latter two baselines were chosen as these methods are *explicitly designed to mitigate demographic bias in federated learning.* To highlight the importance of global communication of fairness information, we also consider `FairFed-Kernel`, which uses `FedAvg` with each local client implementing a local bias mitigation algorithm (similar to (10)). Finally, KFFL is compared against its time delay variant KFFL-TD.

The selection of the hyperparameters for KFFL and its variants is discussed in Appendix B.1. This section includes a discussion of both common hyperparameters (e.g., batch size, learning rate, local epochs, global rounds) and algorithm-specific hyperparameters, such as the feature map size $D$ used in KFFL and KFFL-TD.

We evaluate the performance in two different federated learning settings: IID (independent and identically distributed) and Non-IID. In the IID setting, each client is provided with an equal number of samples and a shared local data distribution Li et al. (2020). In the Non-IID setting, each client has a different distribution of the protected attribute. Specifically, since the protected group $\mathcal{A}$ is binary with attributes $\mathcal{A}_0$ and $\mathcal{A}_1$, half of the clients have 90% of $\mathcal{A}_0$ and 10% of $\mathcal{A}_1$, while the other half has 90% of $\mathcal{A}_1$ and 10% of $\mathcal{A}_0$ Li et al. (2020).

For classification tasks, we used five datasets commonly encountered in group fairness research (Han et al., 2023): `Adult`, `COMPAS`, `Bank`, `ACS`, and `German`. Logistic regression and neural networks are used to evaluate the fairness accuracy trade-off of KFFL (Han et al., 2023). Additional results and details can be found in Appendix D.

When the underlying task is regression, we incorporate additional datasets into our evaluation. Beyond the `Adult` dataset, we also consider the `Law School` and `Communities and Crime` datasets, as utilized in the work Agarwal et al. (2019a). For more details on the datasets refer to Appendix B.2.

**A note on the scope of the experimental evaluation.** Our primary focus in this work is the development and evaluation of a novel kernel-based approach to ensuring group fairness in a principled manner. Similar to prior works in federated learning that do not impose privacy constraints (Crawshaw & Liu, 2024; Cho et al., 2023; Gu et al., 2021; Malinovsky et al., 2023; Eichner et al., 2019), we use no privacy-preserving mechanisms in this study. The integration of privacy-preserving protocols such as DP-SGD (Chua et al., 2024) or secure aggregation (Bonawitz et al., 2017) is left as future work. Furthermore, we consider *full client participation*, consistent with approaches such as `FairFed` (Ezzeldin et al., 2023), the `MinMax` algorithm proposed by Papadaki et al. (2022), and other works in federated learning (Zhang et al., 2023; Li et al., 2023a;b; Zakerinia et al., 2023; Huang et al., 2022). Finally, as wall-clock time differences may be sensitive to differences in the implementation efficiency of our methods versus the baseline methods, our experimental evaluation focuses on communication costs rather than wall-clock time. This focus is justified by the fact that the performance of federated learning techniques is typically communication-bound.

## 6.1 Performance Evaluation of KFFL

Table 1 compares the *communication rounds* required by our proposed methods, KFFL and its time-delay variant KFFL-TD see (C), with baseline methods `FedAvg` and `FairFed` Ezzeldin et al. (2023). While KFFL

Table 1: Communication measured in terms of the number of rounds required for one global update using KFFL, KFFL-TD, `FedAvg`, and `FairFed`. Algorithms that incorporate fairness, such as `FairFed`, require a similar number of communication rounds as our methods KFFL and KFFL-TD.

| Method | Rounds of Communication |
|---|---|
| KFFL | 3 |
| KFFL-TD (Time Delay Variant) | 2 |
| FairFed (Ezzeldin et al., 2023) | 3 |
| FedAvg (McMahan et al., 2017) | 1 |

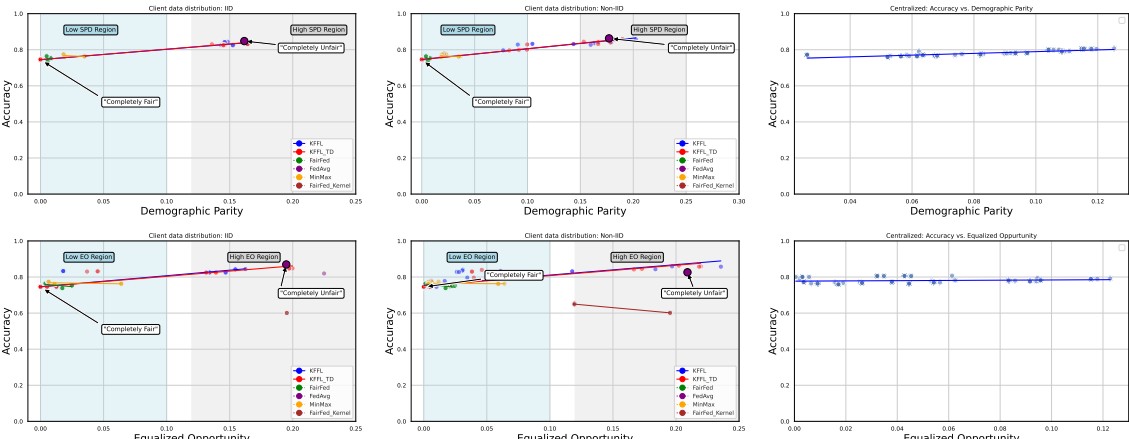

Figure 3: Accuracy versus Demographic Parity (DP) and Equalized Oppurtunity (EO) for KFFL and the baselines under IID and Non-IID conditions on the `ADULT` test dataset. Each point represents a different fairness weight $\lambda$ ranging from 20.00 to 1000.00 for both KFFL and KFFL-TD. The blue region denotes **higher levels of group fairness** (low SPD and EO), while the gray region indicates **lower levels of group fairness** (high SPD and EO).

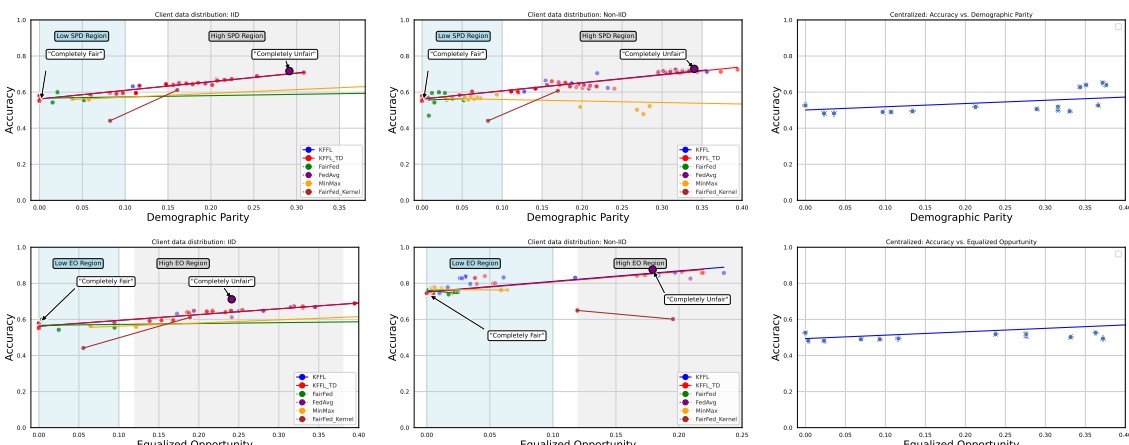

Figure 4: Accuracy versus Demographic Parity (DP) and Equal Opportunity (EO) for KFFL and its baselines under IID and Non-IID conditions on the `COMPAS` test dataset. Each point represents a different fairness weight $\lambda$ ranging 0.01 to 123.16 for both KFFL and KFFL-TD. The blue region denotes higher levels of group fairness (low SPD and EO disparities), while the gray region indicates lower levels of group fairness (high SPD and EO disparities)

requires a similar number of communication rounds as `FairFed`, KFFL-TD reduces the number of rounds needed for a global model update by using stale fairness gradients, thereby lowering the total communication rounds needed per update.

Existing fair classifiers and regressors that balance fairness and accuracy typically optimize for a single point on the trade-off curve (see Figure 4 and Figure 3), whereas KFFL and KFFL-TD explore the trade-offs between accuracy and group fairness more thoroughly as the fairness hyperparameter $\lambda$ is swept. This suggests that the baseline algorithms may lead to suboptimal outcomes when practitioners employ them to achieve specific trade-offs between accuracy and group fairness, while KFFL and KFFL-TD are more suitable for achieving more control over the achieved trade-off.

For the classification tasks, we evaluate the models' test accuracy and fairness metrics, focusing on SPD, (1), and EO, (2), across five datasets: `BANK`, `ACS`, `COMPAS`, `ADULT`, and `GERMAN`. Each dataset considers a single

protected binary sensitive attribute. For example, in `COMPAS`, the protected attribute is race (black/white), while in `ADULT`, it is sex (male/female). This subsection presents results for the logistic regression model, with the Appendix D providing similar results on the performance of the neural network model.

KFFL and its baselines are compared under both IID and Non-IID conditions on the `COMPAS` test dataset in Figure 4. The blue region, referred to as the 'Low SPD Region' and 'Low EO Region,' represents low statistical parity discrepancy and low equalized opportunity discrepancy. These regions correspond to higher levels of group fairness. In contrast, the gray region, labeled as the 'High SPD Region' and 'High EO Region,' indicates lower levels of group fairness.

Points labeled "completely fair" indicate trade-off points where the model achieves no statistical parity or equalized opportunities discrepancies. In contrast, points labeled "completely unfair" represent the performance of the standard `FedAvg` model, which is trained without any fairness objective (i.e., $\lambda = 0$ in the distributed setting of (10)). Each point reflects a different fairness weight $\lambda$, ranging from 20.00 to 1000.00 for both KFFL and KFFL-TD using the `ADULT` dataset and from 0.01 to 123.16 for the `COMPAS` dataset. More details on the choice of $\lambda$ for the other datasets is given in Appendix B.5. As $\lambda$ increases beyond this range, we observe non-optimal points to the right of the "completely unfair" point.

It can be seen that other baselines, such as `FairFed` and `MinMax`, do not produce a smooth trade-off between accuracy and fairness. For `FairFed` (Ezzeldin et al., 2023), the tradeoff is controlled by a parameter $\beta$ called the "fairness budge", which varies from 0.1 to 5, based on the recommendations in Ezzeldin et al. (2023). Optimal performance of the `FairFed` baseline requires a local debiasing mechanism, for which we use the `FairBatch` algorithm (Roh et al., 2020).

For `MinMax`, a "global adversary rate" Papadaki et al. (2022) is used to control the reduction of expected loss for the worst-performing demographic. To explore the accuracy-fairness tradeoff, we varied this parameter from 0.001 to 0.1 based on the settings in that work.

In the right column, the `Centralized` method refers to a non-distributed data setting, corresponding to Equation 10, where the full kernel is used as a regularizer. Similarly, Figure 3 provides an evaluation for the `ADULT` *test* dataset. Each point represents a different fairness weight $\lambda$, ranging from 0 to 0.01 for both KFFL and KFFL-TD.

The `Centralized` version for each dataset illustrates that using the full kernel allows for a clear tradeoff between accuracy and fairness. The federated KFFL algorithm preserves this trade-off in Figures 4 and 3. Specifically, KFFL and KFFL-TD maintain competitive accuracy (approximately 0.85 to 0.78) on the `ADULT` dataset and between 0.6 to 0.7 on the `COMPAS` dataset, *across different fairness regions*. In contrast, `FairFed` and `MinMax` perform similarly to our methods in regions with lower levels of group fairness but significantly underperform in regions with higher levels of group fairness, often failing to provide any trade-off points in the areas identified as high fairness regions.

The smooth exploration of the achievable accuracy–fairness trade-offs by KFFL and KFFL-TD persists even in the more challenging Non-IID setting. Unlike `FairFed` and `MinMax`, which require extensive tuning for specific evaluation metrics, KFFL and KFFL-TD **perform well without metric-specific hyper-parameter optimization**. However, the choice of the fairness weight hyperparameter $\lambda$ is crucial for effectively utilizing KFFL and KFFL-TD. Increasing the fairness weight $\lambda$ beyond a certain threshold can push the model's performance towards lower levels of group fairness, as shown in Figures 4 and 3.

Figures 3 and 4 also illustrate the tradeoff achieved when KHSIC is applied as a local fairness regularizer in the method `FairFed-Kernel`. In these methods, each client independently solves (10) without global communication of fairness gradients. The results clearly show that local debiasing methods fail to achieve a tradeoff comparable to the centralized approach, where distributional information from across all clients is taken into consideration using (10). These findings underscore the importance of the principled strategies employed by KFFL and KFFL-TD.

Table 2: Communication costs of KFFL and KFFL-TD, relative to `FedAvg`. Here, $\varepsilon \triangleq D^2/|\boldsymbol{\omega}|$. The KFFL and KFFL-TD algorithms incur additional cost due to the exchange of fairness information, however, the use of RFMs ensures that this additional cost is vanishing when the model size $|\boldsymbol{\omega}|$ is large.

| Method | Uplink | Downlink |
|---|---|---|
| KFFL | $2 + \varepsilon$ | $2 + \varepsilon$ |
| KFFL-TD (Time Delay Variant) | $2 + \varepsilon$ | $2 + \varepsilon$ |
| Fair Fed (Ezzeldin et al., 2023) | 1 | 1 |
| FedAvg (McMahan et al., 2017) | 1 | 1 |

## 6.2 The communication cost of KFFL

KFFL and KFFL-TD incur additional communication overhead over `FedAvg`, due to the exchange of the parameters needed to compute the fairness gradient. Table 2 compares the relative communication costs (per client, per iteration) for the `FedAvg` and KFFL algorithms. Let $|\boldsymbol{\omega}|$ denote the size of the model. In each iteration, `FedAvg` incurs uplink (client-to-server) and downlink (server-to-client) communication costs of $|\boldsymbol{\omega}|$, the number of parameters in the model. In contrast, KFFL requires the client to transmit $\boldsymbol{\omega}_t^i, \Phi^i(\boldsymbol{\omega}_t)$, and $\mathbf{g}^i(\boldsymbol{\omega}_t)$ (see Algorithm 2), resulting in an uplink cost of $2|\boldsymbol{\omega}| + D^2$, and the server to transmit $\boldsymbol{\omega}_t, \Lambda(\boldsymbol{\omega}_t)$, and $\boldsymbol{\omega}_{t+1/2}$, incurring a similar downlink cost of $2|\boldsymbol{\omega}| + D^2$ (see Algorithm 3). Notably, $D^2$ is often smaller than $|\boldsymbol{\omega}|$, particularly for large models. KFFL-TD (Appendix C) incurs comparable communication costs but reduces the number of communication rounds per iteration (see Figure 7).

To quantify this overhead, Table 2 defines $\varepsilon \triangleq \frac{D^2}{|\boldsymbol{\omega}|}$, where $D$ is the dimension of the RFMs. This results in total uplink and downlink communication costs of $2 + \varepsilon$ per iteration for KFFL, compared to 1 for nonfair methods like `FedAvg`. Since $\varepsilon$ is relatively small when $D^2 \ll |\omega|$, the additional communication cost becomes negligible in practice for large models. Consequently, the overall communication costs of KFFL and KFFL-TD remains approximately two times that of `FedAvg`, due to the additional exchange of model parameters, while providing the benefits of ensuring learning a model that is fair with respect to different demographic groups.

`FairFed` also exchanges local and global accuracy and fairness metrics during each global update, but we exclude these scalar quantities from the comparison for simplicity in Table 2.

## 6.3 Additional examples of KFFL trade-offs

Table 3 highlights selected trade-off points for additional datasets, including `BANK`, `ACS`, and `GERMAN`, alongside the previously analyzed `COMPAS` dataset. The models were trained under the Non-IID setting, and the trade-off points were computed based on performance on the test dataset averaged over three runs.

To showcase the benefits of our approach, we selected specific trade-off points. For the `COMPAS` dataset, if the desired accuracy is around 60%, similar to the performance of `FedAvg`, KFFL and KFFL-TD achieve a small SPD gap of 0.06. In contrast, the baselines achieve a similar SPD gap but with an approximately 5% drop in accuracy. A similar trend is evident in the `ACS` dataset, where KFFL-TD provides a trade-off point of relatively high 80% accuracy for a low SPD gap. For the baselines, a comparable fairness point results in a 22% drop in accuracy.

Additionally, KFFL is robust across different evaluation metrics. On the `BANK` dataset, KFFL and KFFL-TD offer a trade-off point of 90% accuracy with SPD gaps of 0.05 and 0.04, respectively. In comparison, `FairFed` achieves 0.00 SPD gap with 88% accuracy but as a consequence exhibits a high EO gap of 0.16, as all methods were optimized for reducing SPD in our experiments. This underscores the sensitivity of the `FairFed` method to the choice of the evaluation metric. Additionally, `MinMax` shows a high SPD gap but low EO gap, indicating that depending on the dataset one evaluation metric may be favored over the other with this approach. In contrast, KFFL and KFFL-TD provide a trade-off point that reduces unfairness across both evaluation metrics for this dataset.

| | Method | COMPAS | BANK | ACS | GERMAN |
|---|---|---|---|---|---|
| Acc. (↑) | FedAvg | $61.13 \pm 1.25$ | $91.21 \pm 1.10$ | $81.14 \pm 1.20$ | $72.50 \pm 1.15$ |
| | KFFL | $60.30 \pm 1.30$ | $90.23 \pm 1.25$ | $79.12 \pm 1.35$ | $72.50 \pm 1.10$ |
| | KFFL-TD | $59.51 \pm 1.45$ | $90.78 \pm 1.20$ | $81.12 \pm 1.05$ | $72.50 \pm 1.40$ |
| | FairFed/FairBatch | $55.47 \pm 1.50$ | $88.05 \pm 1.30$ | $58.77 \pm 1.20$ | $30.00 \pm 1.25$ |
| | FairFed-Kernel | $44.13 \pm 1.55$ | $88.77 \pm 1.15$ | $58.77 \pm 1.05$ | $70.00 \pm 1.35$ |
| | MinMax | $56.68 \pm 1.40$ | $64.93 \pm 1.20$ | $58.77 \pm 1.25$ | $70.00 \pm 1.10$ |
| SPD (↓) | FedAvg | $0.16 \pm 0.02$ | $0.23 \pm 0.01$ | $0.08 \pm 0.04$ | $0.19 \pm 0.03$ |
| | KFFL | $0.06 \pm 0.01$ | $0.05 \pm 0.03$ | $0.04 \pm 0.02$ | $0.00 \pm 0.04$ |
| | KFFL-TD | $0.05 \pm 0.02$ | $0.06 \pm 0.01$ | $0.00 \pm 0.03$ | $0.00 \pm 0.02$ |
| | FairFed/FairBatch | $0.05 \pm 0.03$ | $0.00 \pm 0.02$ | $0.00 \pm 0.01$ | $0.00 \pm 0.03$ |
| | FairFed-Kernel | $0.08 \pm 0.02$ | $0.00 \pm 0.03$ | $0.00 \pm 0.02$ | $0.00 \pm 0.01$ |
| | MinMax | $0.03 \pm 0.01$ | $0.21 \pm 0.02$ | $0.00 \pm 0.03$ | $0.00 \pm 0.02$ |
| EO (↓) | FedAvg | $0.23 \pm 0.02$ | $0.16 \pm 0.01$ | $0.04 \pm 0.02$ | $0.02 \pm 0.03$ |
| | KFFL | $0.07 \pm 0.01$ | $0.04 \pm 0.02$ | $0.02 \pm 0.03$ | $0.00 \pm 0.02$ |
| | KFFL-TD | $0.08 \pm 0.02$ | $0.05 \pm 0.01$ | $0.03 \pm 0.02$ | $0.00 \pm 0.03$ |
| | FairFed/FairBatch | $0.09 \pm 0.02$ | $0.16 \pm 0.03$ | $0.00 \pm 0.01$ | $0.00 \pm 0.02$ |
| | FairFed-Kernel | $0.05 \pm 0.03$ | $0.00 \pm 0.02$ | $0.00 \pm 0.01$ | $0.00 \pm 0.03$ |
| | MinMax | $0.03 \pm 0.02$ | $0.02 \pm 0.01$ | $0.00 \pm 0.03$ | $0.00 \pm 0.02$ |

Table 3: Selected fairness–accuracy trade-off points for the BANK, ACS, GERMAN, and COMPAS datasets in non-IID settings. The results are averaged over three runs on the test sets. Lower SPD and EO gaps indicate higher fairness. KFFL and KFFL-TD provide superior trade-off points compared to baselines.

## 6.4 KFFL provides fair trade-off points for regression tasks

| Fairness hyperparameter $\lambda$ | RMSE ↓ | KS distance ↓ |
|---|---|---|
| 0.00 | $0.49088 \pm 0.009294$ | $0.38293 \pm 0.016433$ |
| 0.01 | $0.49101 \pm 0.009303$ | $0.39744 \pm 0.027059$ |
| 0.10 | $0.49053 \pm 0.009130$ | $0.37377 \pm 0.030239$ |
| 1.00 | $0.49071 \pm 0.009039$ | $0.38429 \pm 0.053008$ |
| 500.00 | $40.32616 \pm 48.938383$ | $0.27088 \pm 0.139738$ |

Table 4: RMSE and the KS distance with standard deviations for 5 runs of KFFL for the Adult dataset in the IID case. Points with lower RMSE and lower KS distance are preferable.

| Fairness hyperparameter $\lambda$ | RMSE ↓ | KS distance ↓ |
|---|---|---|
| 0.00 | $0.810188 \pm 0.002067$ | $0.439208 \pm 0.175972$ |
| 0.01 | $0.810068 \pm 0.002152$ | $0.424060 \pm 0.164314$ |
| 0.10 | $0.809940 \pm 0.002189$ | $0.487244 \pm 0.139673$ |
| 1.00 | $0.810052 \pm 0.002201$ | $0.200248 \pm 0.170934$ |
| 5.00 | $0.810100 \pm 0.002022$ | $0.212672 \pm 0.065094$ |
| 100.00 | $31.159116 \pm 19.366019$ | $0.184708 \pm 0.081729$ |

Table 5: RMSE and the KS distance with standard deviations for 5 runs of KFFL for the Law School dataset in the Non-IID case. Points with lower RMSE and lower KS distance are preferable.

We evaluate KFFL with a linear regression model, measuring predictive accuracy by the root mean square error (RMSE) and group fairness by the KS distance (Eq. 6); lower values are better for both metrics.

Table 4 reports performance under increasing fairness weights $\lambda$ in the IID setting on the Adult dataset. As $\lambda$ grows, the fairness term gains influence. For small to moderate weights ($\lambda \leq 1$) the KS distance falls by roughly 15%, while RMSE is statistically unchanged. However, once the weight is large enough ($\lambda > 1$) the expected accuracy–fairness trade-off *does* materialize: RMSE increases, yet KS continues to decline.

An analogous pattern appears in the Non-IID setting on the Law School dataset (Table 5).

Additional results in the Appendix— Tables 15, 13, 11, 12, 14, and 9— extend these findings to the Law School, Communities and Crime, and Adult datasets. These results demonstrate the effectiveness

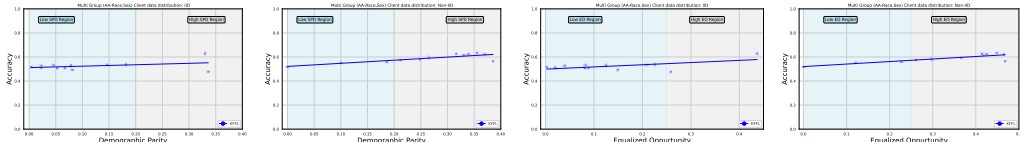

Figure 5: **Intersectional (Race × Sex) fairness on COMPAS.** Accuracy under KFFL versus the multi-attribute SPD and multi-attribute EO gaps. Markers sweep the fairness weight $\lambda \in \{0, 1000, 1100, \dots, 2000\}$; larger $\lambda$ pushes the model from the low fairness region towards the high fairness region. Groups are defined by the Cartesian product *{African-American binary indicator}×{Male,Female}*, yielding $K = 4$ intersectional categories. The two left panels show the trade-offs between the SPD gap and accuracy in the IID and Non-IID settings, while the right half shows the trade-offs with the EO gap under the IID and Non-IID settings.

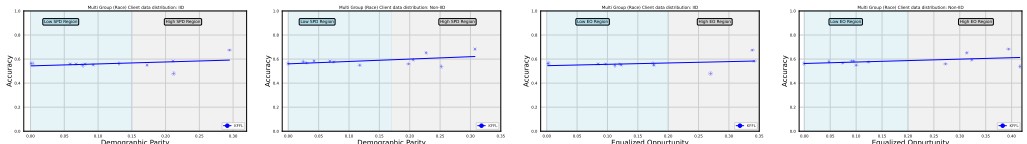

Figure 6: **Multi-racial fairness on COMPAS.** These panels follow the same layout as in Figure 5, but with the sensitive attribute "race" taking values over the full set {African-American, Caucasian, Hispanic, Asian, Native American, Other} ($K = 6$). The transition from the low fairness region to the high fairness region as the fairness hyperparameter $\lambda$ increases illustrates the trade-off between accuracy and multi-group SPD and multi-group EO gaps.

of KFFL in balancing accuracy and fairness in regression tasks by appropriately tuning the fairness hyperparameter $\lambda$.

## 6.5 KFFL yields favourable trade-offs in settings with multiple sensitive attributes

We demonstrate the capability of KFFL to achieve fairness-accuracy trade-offs when there are multiple sensitive attributes by considering (i) an intersectional attribute *race × sex* that takes $K = 4$ values, and (ii) the full racial spectrum recorded in COMPAS, which takes $K = 6$ values.[5] Fairness is measured by the max–min gaps $\text{SPD}_{\text{multi}}$ and $\text{EO}_{\text{multi}}$ ((3)–(4)).

Figures 5 and 6 plot the model accuracy in these two experimental setups, versus the fairness gaps while sweeping the fairness hyperparameter $\lambda \in \{0, 1000, 1100, \dots, 2000\}$ under IID and non-IID federated settings. In every case the results trace a smooth trade-off between fairness and accuracy: increasing $\lambda$ moves the operating point from the high-gap zone into the low-gap zone while keeping the accuracy in the range $[0.60, 0.72]$.

We note that when $\lambda = 1600$ in the IID setting of the intersectional experiment, the model attains $\text{SPD}_{\text{multi}} < 0.05$ and $\text{EO}_{\text{multi}} < 0.05$ with an accuracy of around 0.65. This is a substantial improvement over the unregularised model ($\lambda = 0$), in which $\text{SPD}_{\text{multi}} \approx 0.30$ and $\text{EO}_{\text{multi}} \approx 0.40$. A comparable pattern emerges for the multiple-race experiment, showing that KFFL scales gracefully from binary to genuinely multi-group fairness.

The orthogonal random-feature (ORF) Yu et al. (2016) map utilized in our implementation of KFFL requires $D \geq 2k$, where $k$ is the number of one-hot columns, to satisfy the orthogonality condition. The earlier binary-attribute experiments used $D=10$, but the richer group structure in these multiple sensitive attribute settings demands slightly larger $D$: we set $D=16$ for the intersectional setting ($k=7$) and $D=24$ for the six-race setting ($k=11$). Further increases to $D$ produced no significant changes in either accuracy or fairness, confirming that a simple rule of thumb of $D \approx 2k$ is adequate even in the multiple sensitive attribute regime.

---

[5]Intersectional groups: $(\text{AA} = 1, \text{M})$, $(\text{AA} = 1, \text{F})$, $(\text{AA} = 0, \text{M})$, $(\text{AA} = 0, \text{F})$; racial groups: African-American, Caucasian, Hispanic, Asian, Native American, Other.

# 7 Conclusions

This work introduces a systematic approach for training group-fair machine learning models in a federated setting by leveraging KHSIC as a fairness regularizer to capture complex, non-linear dependencies between model outputs and sensitive attributes. The proposed method, KFFL, significantly reduces the communication and computation costs of a naive implementation by employing random feature maps and a novel federated proximal gradient algorithm, `FedProxGrad`, which accommodates the non-convexity of both the data-fitting term and the fairness regularizer.

Experimental results demonstrate that KFFL performs robustly across diverse client data distributions and standard datasets commonly used to evaluate fair learning methods. It achieves strong performance in both regression and classification tasks by more thoroughly exploring the trade-offs between fairness and accuracy compared to existing baselines. In exchange for this flexibility, KFFL and KFFL-TD incur about twice the communication cost of `FedAvg`, and respectively require two and one more round of communication per iteration in comparison to `FedAvg`. We leave it up to practitioners to determine, based on the requirements of their use case, whether the ability to better explore the accuracy-fairness trade-offs justify this additional cost.

**Limitations and Future Work**

KFFL effectively enforces statistical parity in a principled manner. Expanding the method to address other notions of group fairness— e.g. by utilizing conditional variants of the KHSIC to ensure equalized odds—, would enhance its versatility. Also, the current framework assumes full client participation, making it less suitable for scenarios with partial participation or privacy constraints, and requires two additional rounds of communication over `FedAvg` to compute the fairness gradient in order to guarantee convergence. To address these challenges, future work can incorporate differentially private gradient estimation techniques and develop modifications to KFFL and its analyses to support partial participation and to justify the use of stale gradient information in KFFL-TD, which requires only one additional round of communication over `FedAvg`. Additionally, the convergence rate of FEDPROXGRAD is influenced by the variance $\sigma^2$ of the stochastic gradient estimate for the fairness regularizer, which imposes a fixed floor on the convergence of the method to a stationary point of the fairness-regularized learning objective, and the use of the dissimilarity condition. Future work can overcome these challenges by exploring variance reduction techniques to mitigate the impact of $\sigma^2$ on convergence rates and alternative analysis strategies, such as the approach taken in Yuan & Li (2022), to eliminate the need for the bounded dissimilarity assumption.

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

# A    Appendix

## A.1    Related Works

Methods for ensuring fairness within centralized machine learning are typically categorized into three distinct groups: pre-processing, in-processing, and post-processing Mehrabi et al. (2021). In federated learning, bias mitigation methods predominantly fall into in-processing approaches, although some work has also been done in post-processing methods.

For in-processing methods of bias mitigation in federated learning, Ezzeldin et al. (2023) is notable for its versatility and compatibility with various *local* bias mitigation techniques.In this approach, clients convey their localized fairness metrics to the server, which then optimizes weighting coefficients to minimize the contribution of the poorest-performing client with respect to a chosen fairness metric. Papadaki et al. (2022) optimize model performance on the worst-performing demographic by adopting a minimax optimization framework. Salazar et al. (2022) propose a fairness-aware momentum-based method to address bias in federated learning. The approach in Mehrabi et al. (2022) strives for fair federated learning but requires the server to maintain a validation dataset. Zeng et al. (2021) address the challenge of bias mitigation in federated learning through a bi-level optimization problem; their analysis predominantly pertains to specific loss functions. Pentyala et al. (2022) consider post-processing and pre-processing approaches to ensuring fairness. Cui et al. (2021) require the clients to achieve Pareto optimality with respect to both fairness and accuracy.

Other variants of group fairness have been explored in the federated setting. Hu et al. introduce the concept of bounded group loss as a facet of group fairness in federated learning, although their work does not specifically develop algorithms targeting bias mitigation. Chang & Shokri (2023) analyze how bias within participating clients can propagate during the training process but do not propose methods aimed at explicitly addressing group fairness.A comprehensive summary of additional approaches in group fairness federated learning is provided in Table 3 of Salazar et al. (2024). For baseline comparisons, we limit our focus to most **published works in group fairness federated learning** except the recent Wang et al. (2023).

Significant related works from the centralized setting that aim to ensure fair models include Pérez-Suay et al. (2017), which leverages the (non-kernel) Hilbert-Schmidt Independence Criterion (HSIC) to promote the learning of fair kernel machines, and Baharlouei et al. (2019a), which incorporates the Rényi correlation as a regularization term to achieve statistically fair models.

## A.2    Proof of Theorem 1

In the following, $\mathbb{E}_t[\cdot] = \mathbb{E}[\cdot \,|\, \boldsymbol{\omega}_t]$ denotes the expectation conditioned on all sources of randomness in the algorithm up to and including the calculation of $\boldsymbol{\omega}_t$, and $\mathbb{E}_i[\cdot] = \frac{1}{N}\sum_{i=1}^{N}[\cdot]$ denotes the average of a quantity over the clients.

The computations are complicated by the presence of a composite objective and stochasticity in our estimate of $\nabla\psi(\boldsymbol{\omega})$, but the conceptual outline of the proof of the convergence rate follows that of the proof of the convergence rate for FedProx in Li et al. (2020). Namely,

- First, we establish that the distance between consecutive iterates, $\|\boldsymbol{\omega}_{t+1}-\boldsymbol{\omega}_t\|_2$, is upper bounded by the quantity $\mathbb{E}_i\|\nabla\ell^i(\boldsymbol{\omega}_t) + \nabla\psi(\boldsymbol{\omega}_t)\|_2$, and use the bounded dissimilarity condition to upper bound the latter by a multiple of the composite objective gradient at $\boldsymbol{\omega}_t$, $\|\nabla F(\boldsymbol{\omega}_t)\|_2^2$, plus a term due to noise.

- Next, we use this result and the smoothness of the composite objective to establish that after one round of the algorithm, the composite objective satisfies $\mathbb{E}_t F(\boldsymbol{\omega}_{t+1}) \leq F(\boldsymbol{\omega}_t) - \alpha\mathbb{E}_t\|\nabla F(\boldsymbol{\omega}_t)\|_2^2 +$ (vanishing terms) + (noise).

- We conclude that if the step-size $\alpha$ is chosen appropriately, then the objective decreases in expectation at each iteration, up to the noise level. A standard argument with Jensen's inequality and telescoping sums delivers the claimed convergence rate.

*Proof.* We flesh out the preceeding outline.

**Iterate proximity** To bound the iterate proximity, begin by introducing the local exact minimizer,

$$\widehat{\boldsymbol{\omega}}_{t+1}^i = \operatorname{argmin}_{\boldsymbol{\omega}} \ell_t^i(\boldsymbol{\omega}) := \ell^i(\boldsymbol{\omega}) + \frac{1}{2\alpha}\|\boldsymbol{\omega} - \boldsymbol{\omega}_{t+1/2}\|_2^2.$$

The data-fitting term $\ell^i$ is $L_-$-weakly convex and the quadratic regularizer is $\frac{1}{\alpha}$-strongly convex, so the local objective $\ell_t^i$ is $\mu$ strongly convex for $\mu = \frac{1}{\alpha} - L_-$.

The $\mu$-strong convexity of $\ell_t^i$ implies that the iterate distance between $\boldsymbol{\omega}_t$ and $\boldsymbol{\omega}_{t+1}^i$ can be estimated using the size of the gradient of $\ell_t^i$ at those models:

$$\|\boldsymbol{\omega}_{t+1}^i - \boldsymbol{\omega}_t\|_2 \le \|\boldsymbol{\omega}_{t+1}^i - \widehat{\boldsymbol{\omega}}_{t+1}^i\|_2 + \|\boldsymbol{\omega}_t - \widehat{\boldsymbol{\omega}}_{t+1}^i\|_2$$
$$\le \frac{1}{\mu}\left[\|\nabla\ell_t^i(\boldsymbol{\omega}_{t+1}^i)\|_2 + \|\nabla\ell_t^i(\boldsymbol{\omega}_t)\|_2\right].$$

Employing the $\gamma$-suboptimality of $\boldsymbol{\omega}_{t+1}^i$ to estimate the size of $\|\nabla\ell_t^i(\boldsymbol{\omega}_{t+1}^i)\|_2$ refines this estimate to

$$\|\boldsymbol{\omega}_{t+1}^i - \boldsymbol{\omega}_t\|_2 \le \frac{1+\gamma}{\mu}\|\nabla\ell_t^i(\boldsymbol{\omega}_t)\|_2.$$

We note that

$$\nabla\ell_t^i(\boldsymbol{\omega}_t) = \nabla\ell^i(\boldsymbol{\omega}_t) + \frac{1}{\alpha}(\boldsymbol{\omega}_t - \boldsymbol{\omega}_{t+1/2}) = \nabla\ell^i(\boldsymbol{\omega}_t) + \mathbf{g}_t,$$

and consequently

$$\|\boldsymbol{\omega}_{t+1}^i - \boldsymbol{\omega}_t\|_2 \le \frac{1+\gamma}{\mu}\left[\|\nabla\ell^i(\boldsymbol{\omega}_t) + \nabla\psi(\boldsymbol{\omega}_t)\|_2 + \|\nabla\psi(\boldsymbol{\omega}_t) - \mathbf{g}_t\|_2\right].$$

Using Jensen's inequality delivers

$$\mathbb{E}_t\|\boldsymbol{\omega}_{t+1} - \boldsymbol{\omega}_t\|_2^2 \le \mathbb{E}_t\left[\mathbb{E}_i\|\boldsymbol{\omega}_{t+1}^i - \boldsymbol{\omega}_t\|_2^2\right]$$
$$\le 2\left(\frac{1+\gamma}{\mu}\right)^2 \mathbb{E}_t\left[\mathbb{E}_i\|\nabla\ell^i(\boldsymbol{\omega}_t) + \nabla\psi(\boldsymbol{\omega}_t)\|_2^2 + \|\nabla\psi(\boldsymbol{\omega}_t) - \mathbf{g}_t\|_2^2\right]$$
$$\le 2\left(\frac{1+\gamma}{\mu}\right)^2 \left[B^2\|\nabla F(\boldsymbol{\omega}_t)\|_2^2 + G^2 + \sigma^2\right]. \tag{22}$$

The last inequality holds because of the bounded dissimilarity condition and the upper bound on the variance of $\mathbf{g}_t$.

Similarly,

$$\mathbb{E}_t\|\boldsymbol{\omega}_{t+1} - \boldsymbol{\omega}_t\|_2 \le \mathbb{E}_t\left[\mathbb{E}_i\|\boldsymbol{\omega}_{t+1}^i - \boldsymbol{\omega}_t\|_2\right]$$
$$\le \frac{1+\gamma}{\mu}\mathbb{E}_t\left[\sqrt{\mathbb{E}_i\|\nabla\ell^i(\boldsymbol{\omega}_t) + \nabla\psi(\boldsymbol{\omega}_t)\|_2^2} + \sqrt{\|\nabla\psi(\boldsymbol{\omega}_t) - \mathbf{g}_t\|_2^2}\right]$$
$$\le \frac{1+\gamma}{\mu}\left[B\|\nabla F(\boldsymbol{\omega}_t)\|_2 + G + \sigma\right]. \tag{23}$$

**Objective Decrease** The *L*-smoothness of the composite objective implies that

$$\mathbb{E}_t F(\boldsymbol{\omega}_{t+1}) \leq \mathbb{E}_t \left[ F(\boldsymbol{\omega}_t) + \langle \nabla F(\boldsymbol{\omega}_t), \boldsymbol{\omega}_{t+1} - \boldsymbol{\omega}_t \rangle + \frac{L}{2} \|\boldsymbol{\omega}_{t+1} - \boldsymbol{\omega}_t\|_2^2 \right] \tag{24}$$

$$= F(\boldsymbol{\omega}_t) - \alpha \|\nabla F(\boldsymbol{\omega}_t)\|_2^2 + \mathbb{E}_t \left[ \langle \nabla F(\boldsymbol{\omega}_t), \underbrace{\boldsymbol{\omega}_{t+1} - (\boldsymbol{\omega}_t - \alpha \nabla F(\boldsymbol{\omega}_t))}_{= \Delta_t} \rangle \right]$$

$$+ \frac{L}{2} \mathbb{E}_t \|\boldsymbol{\omega}_{t+1} - \boldsymbol{\omega}_t\|_2^2.$$

Equation 22 establishes that, up to noise terms, the term $\mathbb{E}_t \|\boldsymbol{\omega}_{t+1} - \boldsymbol{\omega}_t\|_2^2$ scales like $\alpha^2 \|\nabla F(\boldsymbol{\omega}_t)\|_2^2$, because $\mu^{-2}$ is on the order of $\alpha^2$. Now we develop a series of estimates to establish that the quantity $\mathbb{E}_t \langle F(\boldsymbol{\omega}_t), \Delta_t \rangle$ also scales like $\alpha^2 \|\nabla F(\boldsymbol{\omega}_t)\|_2^2$, up to noise terms.

We begin by using the $\gamma$-suboptimality of $\boldsymbol{\omega}_{t+1}^i$ to find a useful expression for $\Delta_t$. In particular, $\gamma$-suboptimality implies that

$$\nabla \ell^i(\boldsymbol{\omega}_{t+1}^i) + \frac{1}{\alpha}(\boldsymbol{\omega}_{t+1}^i - \boldsymbol{\omega}_{t+1/2}) = \nabla \ell^i(\boldsymbol{\omega}_{t+1}^i) + \mathbf{g}_t + \frac{1}{\alpha}(\boldsymbol{\omega}_{t+1}^i - \boldsymbol{\omega}_t)$$

$$= \left( \nabla \ell^i(\boldsymbol{\omega}_{t+1}^i) + \nabla \psi(\boldsymbol{\omega}_t) \right) - \left( \nabla \psi(\boldsymbol{\omega}_t) - \mathbf{g}_t \right) + \frac{1}{\alpha}(\boldsymbol{\omega}_{t+1}^i - \boldsymbol{\omega}_t)$$

$$= \mathbf{e}_{t+1}^i,$$

where $\|\mathbf{e}_{t+1}^i\|_2 \leq \gamma \|\nabla \ell_t^i(\boldsymbol{\omega}_t)\|_2$. Consequently,

$$\boldsymbol{\omega}_{t+1} - \boldsymbol{\omega}_t = \mathbb{E}_i \left[ \boldsymbol{\omega}_{t+1}^i - \boldsymbol{\omega}_t \right]$$

$$= \alpha \mathbb{E}_i \left[ \mathbf{e}_{t+1}^i - \left( \nabla \ell^i(\boldsymbol{\omega}_{t+1}^i) + \nabla \psi(\boldsymbol{\omega}_t) \right) + \left( \nabla \psi(\boldsymbol{\omega}_t) - \mathbf{g}_t \right) \right]$$

and, adding and subtracting terms judiciously yields

$$= \alpha \mathbb{E}_i \left[ \mathbf{e}_{t+1}^i - \nabla F(\boldsymbol{\omega}_{t+1}) + \nabla F(\boldsymbol{\omega}_{t+1}) \right.$$

$$- \left( \nabla \ell^i(\boldsymbol{\omega}_{t+1}^i) + \nabla \psi(\boldsymbol{\omega}_t) \right) + \nabla \psi(\boldsymbol{\omega}_{t+1}^i) - \nabla \psi(\boldsymbol{\omega}_{t+1}^i)$$

$$\left. + \left( \nabla \psi(\boldsymbol{\omega}_t) - \mathbf{g}_t \right) \right]$$

$$= -\alpha \nabla F(\boldsymbol{\omega}_{t+1}) - \alpha \mathbb{E}_i \left[ \left( \nabla \ell^i(\boldsymbol{\omega}_{t+1}^i) + \nabla \psi(\boldsymbol{\omega}_{t+1}^i) - \nabla F(\boldsymbol{\omega}_{t+1}) \right) \right]$$

$$- \alpha \mathbb{E}_i \left[ \nabla \psi(\boldsymbol{\omega}_t) - \nabla \psi(\boldsymbol{\omega}_{t+1}^i) \right] + \alpha \left( \nabla \psi(\boldsymbol{\omega}_t) - \mathbf{g}_t \right) + \alpha \mathbb{E}_i \mathbf{e}_{t+1}^i.$$

It follows that

$$\Delta_t = \boldsymbol{\omega}_{t+1} - \boldsymbol{\omega}_t + \alpha \nabla F(\boldsymbol{\omega}_t)$$

$$= -\alpha \underbrace{\left( \nabla F(\boldsymbol{\omega}_{t+1}) - \nabla F(\boldsymbol{\omega}_t) \right)}_{= \mathbf{t}_1} - \alpha \underbrace{\mathbb{E}_i \left[ \nabla \ell^i(\boldsymbol{\omega}_{t+1}^i) + \nabla \psi(\boldsymbol{\omega}_{t+1}^i) - \nabla F(\boldsymbol{\omega}_{t+1}) \right]}_{= \mathbf{t}_2}$$

$$- \alpha \underbrace{\mathbb{E}_i \left[ \nabla \psi(\boldsymbol{\omega}_t) - \nabla \psi(\boldsymbol{\omega}_{t+1}^i) \right]}_{= \mathbf{t}_3} + \alpha \underbrace{\left( \nabla \psi(\boldsymbol{\omega}_t) - \mathbf{g}_t \right)}_{= \mathbf{t}_4} + \alpha \underbrace{\mathbb{E}_i \mathbf{e}_{t+1}^i}_{= \mathbf{t}_5}.$$

Consider the quantity $\mathbb{E}_t \left[ \langle \nabla F(\boldsymbol{\omega}_t), \Delta_t \rangle \right]$:

$$\mathbb{E}_t \left[ \langle \nabla F(\boldsymbol{\omega}_t), \Delta_t \rangle \right] \leq \alpha \mathbb{E}_t \left[ \|\nabla F(\boldsymbol{\omega}_t)\|_2 \cdot \left( \|\mathbf{t}_1\|_2 + \|\mathbf{t}_2\|_2 + \|\mathbf{t}_3\|_2 + \|\mathbf{t}_4\|_2 + \|\mathbf{t}_5\|_2 \right) \right]$$

Observe that because the composite objective is *L*-smooth,

$$\mathbb{E}_t \left[ \|\mathbf{t}_1\|_2 \right] \leq L \mathbb{E}_t \left[ \|\boldsymbol{\omega}_{t+1} - \boldsymbol{\omega}_t\|_2 \right] \leq L \mathbb{E}_t \left[ \mathbb{E}_i \left\| \boldsymbol{\omega}_{t+1}^i - \boldsymbol{\omega}_t \right\|_2 \right].$$

Similarly, the estimate for $\mathbf{t}_3$ uses the *L*-smoothness of the regularizer:

$$\mathbb{E}_t \left[ \|\mathbf{t}_3\|_2 \right] \leq \mathbb{E}_t \left[ \mathbb{E}_i \left\| \nabla \psi(\boldsymbol{\omega}_t) - \nabla \psi(\boldsymbol{\omega}_{t+1}^i) \right\|_2 \right] \leq L \mathbb{E}_t \left[ \mathbb{E}_i \left\| \boldsymbol{\omega}_t - \boldsymbol{\omega}_{t+1}^i \right\|_2 \right].$$

The $\mathbf{t}_2$ term can also be bounded in terms of the iterate distance:

$$\mathbb{E}_t\left[\|\mathbf{t}_2\|_2\right] \le \mathbb{E}_t\left[\mathbb{E}_i\left\|\nabla\ell^i(\boldsymbol{\omega}_{t+1}^i) + \nabla\psi(\boldsymbol{\omega}_{t+1}^i) - \nabla F(\boldsymbol{\omega}_{t+1})\right\|_2\right]$$
$$\le \mathbb{E}_t\left[\mathbb{E}_i\left\|\nabla\ell^i(\boldsymbol{\omega}_{t+1}^i) + \nabla\psi(\boldsymbol{\omega}_{t+1}^i) - \nabla\ell^i(\boldsymbol{\omega}_{t+1}) - \nabla\psi(\boldsymbol{\omega}_{t+1})\right\|_2\right],$$

where the last equality holds because $\nabla\ell(\boldsymbol{\omega}_{t+1}) = \mathbb{E}_i\nabla\ell^i(\boldsymbol{\omega}_{t+1})$. We use the triangle inequality and the $L$-smoothness of $\psi$ and the functions $\ell^i$ to continue our estimation:

$$\le 2L\mathbb{E}_t\left[\mathbb{E}_i\left\|\boldsymbol{\omega}_{t+1}^i - \boldsymbol{\omega}_{t+1}\right\|_2\right] \le 2L\mathbb{E}_t\left\|\boldsymbol{\omega}_{t+1} - \boldsymbol{\omega}_t\right\|_2 + 2L\mathbb{E}_t\left[\mathbb{E}_i\left\|\boldsymbol{\omega}_t - \boldsymbol{\omega}_{t+1}^i\right\|_2\right]$$
$$\le 4L\mathbb{E}_t\left[\mathbb{E}_i\left\|\boldsymbol{\omega}_t - \boldsymbol{\omega}_{t+1}^i\right\|_2\right].$$

Thus we find that

$$\mathbb{E}_t\left[\|\nabla F(\boldsymbol{\omega}_t)\|_2\cdot\left(\|\mathbf{t}_1\|_2 + \|\mathbf{t}_2\|_2 + \|\mathbf{t}_3\|_2\right)\right] \le 6L\|\nabla F(\boldsymbol{\omega}_t)\|_2\cdot\mathbb{E}_t\left[\mathbb{E}_i\left\|\boldsymbol{\omega}_t - \boldsymbol{\omega}_{t+1}^i\right\|_2\right]$$
$$\le \frac{6L(1+\gamma)}{\mu}\left[B\|\nabla F(\boldsymbol{\omega}_t)\|_2^2 + \|\nabla F(\boldsymbol{\omega}_t)\|_2\cdot(G+\sigma)\right]$$
$$\le \frac{6L(1+\gamma)}{\mu}\left((B+1)\|\nabla F(\boldsymbol{\omega}_t)\|_2^2 + \sigma^2 + G^2\right).$$

The last two inequalities are justified by equation 23 and the fact that $|ab| \le \frac{1}{2}(a^2+b^2)$ for any real numbers $a$ and $b$.

The noise term $\mathbf{t}_4$ is controlled by the variance of the stochastic gradient estimate

$$\mathbb{E}_t\left[\|\mathbf{t}_4\|_2\right] \le \sqrt{\mathbb{E}_t\left\|\nabla\psi(\boldsymbol{\omega}_t) - \mathbf{g}_t\right\|^2} \le \sigma.$$

To control the $\mathbf{t}_5$ term, recall that $\|\mathbf{e}_{t+1}^i\|_2 \le \gamma\|\nabla\ell_t^i(\boldsymbol{\omega}_t)\|_2$. This implies that

$$\mathbb{E}_t\left[\|\mathbf{t}_5\|_2\right] = \mathbb{E}_t\left\|\mathbb{E}_i\mathbf{e}_{t+1}^i\right\|_2 \le \mathbb{E}_t\left[\mathbb{E}_i\left\|\mathbf{e}_{t+1}^i\right\|_2\right]$$
$$\le \gamma\mathbb{E}_t\left[\mathbb{E}_i\left\|\nabla\ell^i(\boldsymbol{\omega}_t) + \mathbf{g}_t\right\|_2\right]$$
$$\le \gamma\mathbb{E}_t\left[\mathbb{E}_i\left\|\nabla\ell^i(\boldsymbol{\omega}_t) + \nabla\psi(\boldsymbol{\omega}_t)\right\|_2 + \left\|\nabla\psi(\boldsymbol{\omega}_t) - \mathbf{g}_t\right\|_2\right]$$
$$\le \gamma\left(B\left\|\nabla F(\boldsymbol{\omega}_t)\right\|_2 + G + \sigma\right).$$

The last inequality holds because of Jensen's inequality, the bounded dissimilarity condition, and the bound on the variance of the stochastic gradient estimate.

From these last two estimates, we find that

$$\mathbb{E}_t\left[\|\nabla F(\boldsymbol{\omega}_t)\|_2\cdot\left(\|\mathbf{t}_4\|_2 + \|\mathbf{t}_5\|_2\right)\right] \le \|\nabla F(\boldsymbol{\omega}_t)\|_2\cdot\sigma + \gamma\left(B\|\nabla F(\boldsymbol{\omega}_t)\|_2^2 + \|\nabla F(\boldsymbol{\omega}_t)\|_2(G+\sigma)\right)$$
$$\le \frac{1}{2}\|\nabla F(\boldsymbol{\omega}_t)\|_2^2 + \frac{1}{2}\sigma^2 + \gamma\left((B+1)\|\nabla F(\boldsymbol{\omega}_t)\|_2^2 + G^2 + \sigma^2\right),$$

and therefore we conclude that

$$\mathbb{E}_t\left[\langle\nabla F(\boldsymbol{\omega}_t),\Delta_t\rangle\right] \le \alpha\frac{6L(1+\gamma)}{\mu}\left((B+1)\|\nabla F(\boldsymbol{\omega}_t)\|_2^2 + G^2 + \sigma^2\right) \tag{25}$$
$$+ \frac{\alpha}{2}\|\nabla F(\boldsymbol{\omega}_t)\|_2^2 + \frac{\alpha}{2}\sigma^2 + \alpha\gamma\left((B+1)\|\nabla F(\boldsymbol{\omega}_t)\|_2^2 + G^2 + \sigma^2\right)$$
$$\le \alpha\cdot c_{\gamma,L,B,\mu}\cdot\left(\|\nabla F(\boldsymbol{\omega}_t)\|_2^2 + G^2 + \sigma^2\right),$$

where, for convenience, we defined

$$c_{\gamma,L,B,\mu} = \left(\frac{1}{2} + \gamma(B+1) + \frac{6L(1+\gamma)(B+1)}{\mu}\right).$$

Finally, consider the squared iterate distance in equation 24. In particular, equation 22 implies that

$$\frac{L}{2}\mathbb{E}_t\|\boldsymbol{\omega}_{t+1} - \boldsymbol{\omega}_t\|_2^2 \leq L\left(\frac{1+\gamma}{\mu}\right)^2\left[B^2\|\nabla F(\boldsymbol{\omega}_t)\|_2^2 + G^2 + \sigma^2\right].$$

Using this estimate and equation 25 in equation 24 gives that

$$\mathbb{E}_t\left[F(\boldsymbol{\omega}_{t+1})\right] \leq F(\boldsymbol{\omega}_t) - \alpha\|\nabla F(\boldsymbol{\omega}_t)\|_2^2 + \mathbb{E}_t\left[\langle\nabla F(\boldsymbol{\omega}_t), \Delta_t\rangle\right] + \frac{L}{2}\mathbb{E}_t\left[\|\boldsymbol{\omega}_{t+1} - \boldsymbol{\omega}_t\|_2^2\right] \tag{26}$$

$$\leq F(\boldsymbol{\omega}_t) - \alpha\|\nabla F(\boldsymbol{\omega}_t)\|_2^2 + c_{\gamma,L,B,\mu,\alpha}\|\nabla F(\boldsymbol{\omega}_t)\|_2^2 + c_{\gamma,L,B,\mu,\alpha}(G^2 + \sigma^2),$$

where, for convenience, we define

$$c_{\gamma,L,B,\mu,\alpha} = \alpha \cdot c_{\gamma,L,B,\mu} + LB^2 \cdot \left(\frac{1+\gamma}{\mu}\right)^2$$

$$= \alpha\left(\frac{1}{2} + \gamma(B+1) + \frac{6L(1+\gamma)(B+1)}{\mu}\right) + LB^2 \cdot \left(\frac{1+\gamma}{\mu}\right)^2$$

Because $\alpha < \frac{1}{2L_-}$,

$$\frac{1}{\mu} = \frac{\alpha}{1 - \alpha L_{-1}} < 2\alpha,$$

and consequently

$$c_{\gamma,L,B,\mu,\alpha} \leq \alpha\left(\frac{1}{2} + \gamma(B+1) + 12\alpha L(1+\gamma)(B+1)\right) + 4\alpha^2 LB^2(1+\gamma)^2.$$

We also choose $\gamma < \frac{1}{8(B+1)}$ and $\gamma < \frac{1}{20}$, which further implies that

$$c_{\gamma,L,B,\mu,\alpha} \leq \alpha\left(\frac{5}{8} + 13\alpha L(B+1)\right) + 5\alpha^2 LB^2,$$

and because $\alpha < \min\left\{\frac{1}{120L(B+1)}, \frac{1}{5LB^2}, \frac{1}{20}\right\}$, in fact

$$c_{\gamma,L,B,\mu,\alpha} \leq \frac{3}{4}\alpha + \alpha^2 \leq \frac{4}{5}\alpha.$$

Thus, we conclude that the expected decrease in the composite objective at each iteration satisfies

$$\mathbb{E}_t\left[F(\boldsymbol{\omega}_{t+1})\right] \leq F(\boldsymbol{\omega}_t) - \frac{\alpha}{5}\|\nabla F(\boldsymbol{\omega}_t)\|_2^2 + \frac{4\alpha}{5}(G^2 + \sigma^2). \tag{27}$$

**Convergence rate** In the remainder of this proof, $\mathbb{E}[\cdot]$ denotes the expectation with respect to all sources of randomness and we take $G = 0$. Using the tower rule of expectations and summing over the first $T$ iterations, we find that

$$\frac{\alpha}{5}\sum_{t=0}^{T-1}\mathbb{E}\left[\|\nabla F(\boldsymbol{\omega}_t)\|_2^2\right] - \frac{4\alpha T}{5}\sigma^2 \leq \sum_{t=0}^{T}\mathbb{E}\left[F(\boldsymbol{\omega}_t) - F(\boldsymbol{\omega}_{t+1})\right]$$

$$= F(\boldsymbol{\omega}_0) - \mathbb{E}\left[F(\boldsymbol{\omega}_{T+1})\right] \leq F(\boldsymbol{\omega}_0) - F^\star.$$

Rearranging terms yields the claimed result:

$$\frac{1}{T}\sum_{t=0}^{T-1}\mathbb{E}\left[\|\nabla F(\boldsymbol{\omega}_t)\|_2^2\right] \leq \frac{F(\boldsymbol{\omega}_0) - F^\star}{\alpha T} + 4\sigma^2.$$

$\square$

### A.3 Proof of Theoram 2

*Proof.* We know from the definition of KHSIC that

$$
\begin{aligned}
\psi(\boldsymbol{\omega}) &= \frac{1}{(n-1)^2} \mathrm{Tr}(\mathbf{K}_s \mathbf{H} \mathbf{K}_f \mathbf{H}) \\
&= \frac{1}{(n-1)^2} \mathbb{E}\left[\mathrm{Tr}(\mathbf{Z}_s \mathbf{Z}_s^\top \mathbf{H} \mathbf{Z}_f \mathbf{Z}_f^\top \mathbf{H})\right] \\
&= \frac{1}{(n-1)^2} \mathbb{E}\left[\mathrm{Tr}(\mathbf{Z}_s^\top \mathbf{H} \mathbf{Z}_f \cdot \mathbf{Z}_f^\top \mathbf{H} \mathbf{Z}_s)\right] \\
&= \frac{1}{(n-1)^2} \mathbb{E}\left[\|\mathbf{Z}_s^\top \mathbf{H} \mathbf{Z}_f\|_F^2\right].
\end{aligned}
$$

The first equality is the definition of the KHISC, and the second holds because the outer products of the random feature maps are the kernel matrices, in expectation. The third holds due to the cyclicity of the trace operator, and the final holds by the definition of the Frobenius norm. The rest of the proof follows Lemma 1 □

### A.4 Proof of Lemma 1:

Consider the following:

*Proof.* The reduced size fairness interaction matrix can be computed efficiently, by noting that

$$
\mathbf{Z}_s^\top \mathbf{H} \mathbf{Z}_f = \mathbf{Z}_s^\top \mathbf{H} \mathbf{H} \mathbf{Z}_f
$$

Thus,

$$
\begin{aligned}
\mathbf{Z}_s^\top \mathbf{H} &= \mathbf{Z}_s^\top \left(\mathbf{I} - \frac{1}{n}\mathbf{1}\mathbf{1}^\top\right) = \mathbf{Z}_s^\top - \boldsymbol{\mu}_s \mathbf{1}^\top, \text{ and} \\
\mathbf{H} \mathbf{Z}_f &= \left(\mathbf{I} - \frac{1}{n}\mathbf{1}\mathbf{1}^\top\right) \mathbf{Z}_f = \mathbf{Z}_f - \mathbf{1}\boldsymbol{\mu}_f^\top.
\end{aligned}
$$

Putting these identities together and using the local paritition of the random feature matrices and their means, we have that

$$
\begin{aligned}
\mathbf{Z}_s^\top \mathbf{H} \mathbf{Z}_f &= \mathbf{Z}_s^\top \mathbf{Z}_f - \boldsymbol{\mu}_s \mathbf{1}^\top \mathbf{Z}_f - \mathbf{Z}_s^\top \mathbf{1} \boldsymbol{\mu}_f^\top + n\boldsymbol{\mu}_s \boldsymbol{\mu}_f^\top \\
&= \mathbf{Z}_s^\top \mathbf{Z}_f - n\boldsymbol{\mu}_s \boldsymbol{\mu}_f^\top \\
&= \sum_{i=1}^m \mathbf{Z}_{s,i}^\top \mathbf{Z}_{f,i} - n \left(\frac{1}{n}\sum_{i=1}^m n_i \boldsymbol{\mu}_{s,i}\right) \left(\frac{1}{n}\sum_{i=1}^m n_i \boldsymbol{\mu}_{f,i}\right)^\top,
\end{aligned} \tag{28}
$$

□

### A.5 Proof of Corollary 1

*Proof.* We use the linearity of expectation to obtain an unbiased approximation of the gradient of the fairness regularizer:

$$
\begin{aligned}
\nabla_{\boldsymbol{\omega}} \psi(\boldsymbol{\omega}) &= \mathbb{E}\, \nabla_{\boldsymbol{\omega}} \tilde{\psi}(\boldsymbol{\omega}) \\
&= \frac{1}{(n-1)^2} \mathbb{E}\, \nabla_{\boldsymbol{\omega}} \|\mathbf{G}(\boldsymbol{\omega})\|_F^2.
\end{aligned}
$$

□

### A.6 Proof of Lemma 2

The stochastic gradient can then be computed in terms of the Jacobians of the local interaction terms $\mathbf{\Omega}_i$ by a simple application of chain rule:

$$\frac{1}{(n-1)^2}\nabla_{\boldsymbol{\omega}}\|\mathbf{G}(\boldsymbol{\omega})\|_F^2 = \frac{2}{(n-1)^2}\mathbf{J_G}(\boldsymbol{\omega})^T\mathbf{G}(\boldsymbol{\omega})$$
$$= \frac{2}{(n-1)^2}\sum_{i=1}^{m}\mathbf{J_{G_i}}(\boldsymbol{\omega})^T\mathbf{G}(\boldsymbol{\omega}).$$

## B  Details of the experimental Evaluation

### B.1  Choice of Hyperparameters

**Common Hyperparameters:** To ensure a fair comparison, all evaluated algorithms utilize a consistent set of common hyperparameters. The batch size is uniformly set to 64. Each algorithm undergoes a total of 10 global training rounds, with each round comprising 5 local epochs on every client. This choice of local epochs ensured that the global model converges within 10 rounds or fewer for all datasets and distribution, allowing us to limit the number of global rounds to 10. The experiments are conducted with 4 clients, a configuration determined based on recent studies. Ezzeldin et al. (2023); Papadaki et al. (2022). The learning rate $\alpha$ was set to 0.01 and Adam Optimizer Kingma & Ba (2014) was used for optimization

**Algorithm Specific Hyperparameters:** Some hyperparameters are specific to the algorithm being used. For example, the **KFFL** and **KFFL-TD** algorithms rely on feature maps **D** to estimate the kernel regularizer. In our experiments, we use the Pyrfm librarypyr to generate random feature maps based on Orthogonal Random Features Yu et al. (2016). The dimensionality of the feature maps used for kernel approximation, denoted as **D**, is set to 10. While higher dimensions also yielded good results, we selected the smallest feature map size that ensured **KFFL** performed effectively..

For other baselines such as Ezzeldin et al. (2023), a tradeoff parameter called "fairness budget" $\beta$ is used to control the effect of reweighing. This tradeoff parameter determines the balance between model accuracy and a specific evaluation metric, by varying $\beta$ from 0.1 to 5 based on the suggestions provided in the paper. However, it should be noted that for the best performance of the Ezzeldin et al. (2023) baseline, a local debiasing mechanism is required. Based on the results from the paper, we used the Roh et al. (2020) algorithm as a local demographic bias mitigation algorithm to compare with our method. Papadaki et al. (2022) use a "global adversary rate" to control how the expected loss over the worst-performing demographic is reduced. To consider an accuracy-fairness tradeoff, we varied this parameter from 0.001 to 0.1.

To enable fine-grained control over the tradeoff between fairness and other performance metrics, our methods incorporate a controllable fairness weight $\lambda$. This weight can be fine-tuned based on the desired tradeoff.More on this in Section B.5

### B.2  Dataset

**Datasets for Classification Task**

- **ADULT**: Becker & Kohavi (1996) is a binary classification dataset that contains up to 14 attributes used in predicting whether an individual would earn an income $\geq 50K$ or $\leq 50K$. The features used in the prediction include continuous attributes such as age, hours per week worked, etc, and discrete attributes including relationship, race, sex, and education. For the purpose of our experimental evaluation, we train a Logistic Regression model Mohri et al. (2019) on this dataset and we consider **sex** as the protected sensitive variable.

- **COMPAS**:Barenstein (2019) is used for predicting criminal recidivism for individuals.The number of samples considered is 6,172 samples and the number of predictive features used in determining

recidivism is 52 including race, age, and previous criminal offenses. For the purpose of our experimental evaluation on this dataset, we consider **race** as the protected sensitive variable evaluated on a Logistic Regression model.

- **BANK**:Data from a Portuguese bank utilized to forecast client subscriptions to term deposits Han et al. (2023).Here we consider **age** as the sensitive attribute and Loan Approval as the target attribute.There are 64 predictive features (including the sensitive) and 41188 target samples.

- **ACS**: From the American Community Survey, utilized for various prediction tasks including income and employment Han et al. (2023). There are 910 (including the sensitive) predictive features and 195665 samples for this dataset. We consider income as the target variable with **sex** as the protected attribute.

- **GERMAN**:Dataon credit applicants from a German bank used for predicting credit risk ratings Han et al. (2023) where the sensitive attribute we consider is **sex** and the target attribute is Credit risk rating. There are 60 (including the sensitive) predictive features and 1000 target samples.

**Datasets for Regression Task**

- `Law School`: Sourced from the Law School Admissions Council's National Longitudinal Bar Passage Study Wightman (1998), this dataset contains 20,649 examples. The task is to predict a student's GPA—normalized to the range $[0, 1]$—using squared loss minimization. Race serves as the protected attribute, categorized as white versus non-white.

- `Communities and Crime`: This dataset comprises socio-economic, law enforcement, and crime statistics from various U.S. communities Redmond & Baveja (2002), totaling 1,994 examples. The objective is to predict the number of violent crimes per 100,000 inhabitants, normalized to $[0, 1]$, through squared loss minimization. The protected attribute is race, defined by whether the community's majority population is white.By including these datasets, we aim to thoroughly evaluate the fair regression estimator's performance across different contexts where fairness with respect to sensitive attributes like race is crucial.

## B.3  Data Distribution on Clients

We explore both the IID (Independent and Identically Distributed) and Non-IID (Non-Independent and Non-Identically Distributed) settings in our evaluation:

- **IID**: In this setting, each client is provided with an equal number of samples and a consistent data distribution for local training Li et al. (2020)

- **Non-IID**: In this setting, each of the clients has different distribution of the protected attribute. Particularly, in our case, since the protected group $\mathcal{A}$ is binary with attributes being $\mathcal{A}_0$ and $\mathcal{A}_1$ , half of the clients have 90 % of $\mathcal{A}_0$ and 10 % of $\mathcal{A}_1$ while the other half has 90 % of $\mathcal{A}_1$ and 10 % of $\mathcal{A}_0$ Li et al. (2020)

## B.4  Models

We consider two distinct types of models for classification and a **linear model for regression**, which clients use for local training. In the fairness literature, Logistic Regression is commonly employed for fair classification tasks Ezzeldin et al. (2023), while linear models are the standard choice for fair regression Chzhen et al. (2020a). Beyond these, we extend our evaluation to include a more complex, non-convex model for classification.

- **Logistic Regression**: This involves a binary logistic regression model with a sigmoid activation function Han et al. (2023)

- **Neural Network**: We also examine the performance of our algorithm using a neural network configuration. This neural network consists of a single hidden layer with 100 neurons and employs ReLU activation, culminating in an output layer.

### B.5 Fairness weights

- **ADULT:** 0.00,20.00, 71.58, 123.16, 174.74, 226.32, 277.89, 329.47, 381.05, 432.63, 484.21, 535.79, 587.37, 690.53, 742.11, 793.68, 845.26, 896.84, 948.42, 1000.00

- **COMPAS:** 0.00, 0.01, 0.10, 20.00, 71.58, 123.16

- **BANK:** 0.00, 0.01, 0.10, 1.00

- **GERMAN:** 0.00, 0.01, 0.10, 1.00

- **ACS:** 0.00, 0.01, 0.10, 1.00

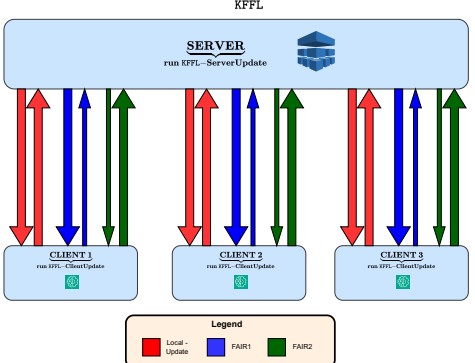 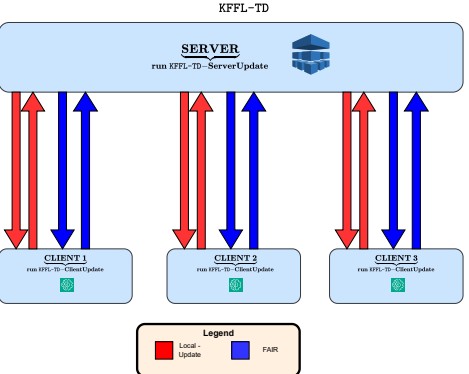

Figure 7: `KFFL` and `KFFL-TD` are illustrated in the figure. Different colors correspond to various segments of communication associated with either the **FAIR1**, **FAIR2** or **Local Update** sub-rounds (see the `KFFL` Algorithm for detail on these flags). For `KFFL-TD` the relevant flags are **FAIR** and **Local Update** (see see the `KFFL-TD` Algorithm for detail on these flags) The direction of the arrows indicate an uplink or downlink communication and the width of each arrowhead highlights the communication cost in each sub-round. Thicker lines indicate higher communication overhead, while thinner lines represent smaller overhead.

## C    KFFL-TD - Kernel Regualarized Fair Learning with time delay

The `KFFL-TD` variant optimizes communication efficiency by incorporating delayed information for the fair term. Assuming the training begins at round $t$ with $t \geq 1$ ; the server transmits the current model Algorithm 5 $\omega_t$ and all global information from the preceding time step $\boldsymbol{\mu}_s(t-1), \boldsymbol{\mu}_f(t-1), \mathbf{G}(\omega_{t-1})$ to the client for local gradient computation. This set of downlink information is denoted as $\Gamma(\omega_t)$. In this case, the server also shares a common seed $\zeta$ to control the randomness in the generation of random feature maps generated. The client also receives a **FAIR** flag, indicating that no data-fitting operation is required. The clients leverage this information to calculate the fair gradient $\mathbf{g}^i(\omega_{t-1})$ (if the global round is not zero) and relevant details $\Phi^i(\omega_t)$ (see Algorithm 4), contributing to the computation of the global interaction $\mathbf{G}(\omega_t)$, $\boldsymbol{\mu}_s(t)$, and $\boldsymbol{\mu}_f(t)$ which will be used in the subsequent round for the fair update (see Algorithm 5) . All of this information is $\Psi^i(\omega_t)$ sent by each client.

With these outdated fair gradients, the server updates the global model see Algorithm 5 and compute the global fairness interaction terms $\mathbf{G}(\omega_t)$,$\boldsymbol{\mu}_s(t)$ to be used by the clients in the next round. To steer the model towards the data-fit direction, the server sends **Local Update** flag instructing the clients perform

---

**Algorithm 4** `KFFL-TD` – Client Update

---

**Input**: $(\mathbf{ROUND}, ..)$

1: **if ROUND** = **FAIR** **then**
2:    **if** $t \neq 0$ **then**
3:       Clients compute local interaction for gradients using Equation 15 at $\boldsymbol{\omega}_{t-1}$ to get $\boldsymbol{\Omega}_i(\boldsymbol{\omega}_{t-1})$
4:       The clients then compute the local gradient $\mathbf{g}^i(\boldsymbol{\omega}_{t-1})$ using Equation 16
5:    **end if**
6:    Clients compute $\mathbf{M}^i(\boldsymbol{\omega}_t)$ using Equation 14 with random seed $\zeta$ and RFMs (such as ORFMs Yu et al. (2016)) , $\boldsymbol{\mu}_{f,i}(t)$ using Equation 18 and $\boldsymbol{\mu}_{s,i}(t)$ using 17
7:    $\Phi^i(\boldsymbol{\omega}_t) = \{\mathbf{M}^i(\boldsymbol{\omega}_t), \boldsymbol{\mu}_{s,i}(t), \boldsymbol{\mu}_{f,i}(t)\}$
8:    **if** $t \neq 0$ **then**
9:       $\Psi^i(\boldsymbol{\omega}_t) = \{\mathbf{g}^i(\boldsymbol{\omega}_{t-1}), \Phi^i(\boldsymbol{\omega}_t)\}$
10:    **else**
11:       $\Psi^i(\boldsymbol{\omega}_t) = \Phi^i(\boldsymbol{\omega}_t)$
12:    **end if**
13:    **Return**: $\Psi^i(\boldsymbol{\omega}_t)$
14: **else if ROUND** = **Local Update** **then**
15:    Clients do a local update on $\boldsymbol{\omega}_{t+1/2}$ following Equation 21 to get $\boldsymbol{\omega}_{t+1}^i$
16:    **Return**: $\boldsymbol{\omega}_{t+1}^i$
17: **end if**

---

**Algorithm 5** `KFFL-TD` – Server Side

---

1: $\boldsymbol{\omega} = \boldsymbol{\omega}_0$ {This is the initial model}
2: $t \leftarrow 0$
3: **while** $\boldsymbol{\omega}$ not converge **do**
4:    **for all** $i = 1, \ldots, m$ in parallel **do**
5:       Generation of random seed $\zeta$
6:       **if** $t \neq 0$ **then**
7:          $\Gamma(\boldsymbol{\omega}_t) = \{\boldsymbol{\omega}_t, \boldsymbol{\mu}_s(t-1), \boldsymbol{\mu}_f(t-1), \mathbf{G}(\boldsymbol{\omega}_{t-1}), \zeta, t\}$
8:       **else**
9:          $\Gamma(\boldsymbol{\omega}_t) = \{\boldsymbol{\omega}_t, \zeta, t\}$
10:       **end if**
11:       $\Psi(\boldsymbol{\omega}_t) = \mathbf{Client\ Update}(\Gamma(\boldsymbol{\omega}_t), $**FAIR**$)$
12:    **end for**
13:    **if** $t \neq 0$ **then**
14:       $\boldsymbol{\omega}_{t+1/2} \leftarrow \boldsymbol{\omega}_t - \sum_{i=1}^m \mathbf{g}^i(\boldsymbol{\omega}_{t-1})$
15:    **end if**
16:    From $\Psi(\boldsymbol{\omega}_t)$ clients get $\Phi(\boldsymbol{\omega}_t) = \{\Phi^i(\boldsymbol{\omega}_t)\}_{i=1}^m$
17:    From $\Phi(\boldsymbol{\omega}_t)$ compute $\mathbf{G}(\boldsymbol{\omega}_t)$ using Equation 13; $\boldsymbol{\mu}_s(t)$ using Equation 17; and $\boldsymbol{\mu}_f(t)$ using Equation 18
18:    **for all** $i = 1, \ldots, m$ in parallel **do**
19:       **if** $t \neq 0$ **then**
20:          $\boldsymbol{\omega}_{t+1}^i = \mathbf{Client\ Update}(\boldsymbol{\omega}_{t+1/2}, $**Local Update**$)$
21:       **else**
22:          $\boldsymbol{\omega}_{t+1}^i = \mathbf{Client\ Update}(\boldsymbol{\omega}_t, $**Local Update**$)$
23:       **end if**
24:    **end for**
25:    $\boldsymbol{\omega}_{t+1} \leftarrow$ `FedAvg`$(\boldsymbol{\omega}_{t+1}^i)$
26: **end while**

---

local optimization (see Equation 21) and communicate the updated copy to the server ; the server conducts `Fed-Avg` after receiving these locally updated models, resulting in the generation of a the updated model. The process continues till the model $\boldsymbol{\omega}_t$ converges.

# D    Additional Empirical Evaluation of Classification

Table 3 and 7 show the results of `KFFL` and its baselines on Logistic Regression and Neural Network across different datasets.KFFL is most robust across datasets in the **Non-IID** setting and provides greater tradeoffs for accuracy and fairness.

| | Method | Adult | COMPAS | BANK | ACS | GERMAN |
|---|---|---|---|---|---|---|
| Acc. | FedAvg | 84.30 ± 1.41 | 61.13 ± 1.25 | 91.21 ± 1.10 | 81.14 ± 1.20 | 72.50 ± 1.15 |
| | KFFL | 83.14 ± 0.42 | 60.3 ± 1.30 | 90.23 ± 1.25 | 79.12 ± 1.35 | 72.50 ± 1.10 |
| | KFFL-TD | 83.97 ± 1.12 | 59.51 ± 1.45 | 90.78 ± 1.20 | 81.12 ± 1.05 | 72.50 ± 1.40 |
| | FairFed/FairBatch | 76.34 ± 0.1 | 55.47 ± 1.50 | 88.05 ± 1.30 | 58.77 ± 1.20 | 30.00 ± 1.25 |
| | KHSIC-Local | 63.40 ± 0.03 | 44.13 ± 1.55 | 88.77 ± 1.15 | 58.77 ± 1.05 | 70.00 ± 1.35 |
| | MinMax | 76.34 ± 0.1 | 56.68 ± 1.40 | 64.93 ± 1.20 | 58.77 ± 1.25 | 70.00 ± 1.10 |
| SPD | FedAvg | 0.18 ± 0.06 | 0.16 ± 0.02 | 0.23 ± 0.01 | 0.08 ± 0.04 | 0.19 ± 0.03 |
| | KFFL | 0.14 ± 0.02 | 0.06 ± 0.01 | 0.05 ± 0.03 | 0.04 ± 0.02 | 0.00 ± 0.04 |
| | KFFL-TD | 0.16 ± 0.03 | 0.05 ± 0.02 | 0.06 ± 0.01 | 0.00 ± 0.03 | 0.00 ± 0.02 |
| | FairFed/FairBatch | 0.004 ± 0.001 | 0.05 ± 0.03 | 0.00 ± 0.02 | 0.00 ± 0.01 | 0.00 ± 0.03 |
| | KHSIC-Local | 0.34 ± 0.00 | 0.08 ± 0.02 | 0.00 ± 0.03 | 0.00 ± 0.02 | 0.00 ± 0.01 |
| | MinMax | 0.004 ± 0.001 | 0.03 ± 0.01 | 0.21 ± 0.02 | 0.00 ± 0.03 | 0.00 ± 0.02 |
| EO | FedAvg | 0.22 ± 0.03 | 0.23 ± 0.02 | 0.16 ± 0.01 | 0.04 ± 0.02 | 0.02 ± 0.03 |
| | KFFL | 0.12 ± 0.03 | 0.07 ± 0.01 | 0.04 ± 0.02 | 0.02 ± 0.03 | 0.00 ± 0.02 |
| | KFFL-TD | 0.04 ± 0.1 | 0.08 ± 0.02 | 0.05 ± 0.01 | 0.03 ± 0.02 | 0.00 ± 0.03 |
| | FairFed/FairBatch | 0.013 ± 0.001 | 0.09 ± 0.02 | 0.16 ± 0.03 | 0.00 ± 0.01 | 0.00 ± 0.02 |
| | FairFed-Kernel | 0.12 ± 0.00 | 0.05 ± 0.03 | 0.00 ± 0.02 | 0.00 ± 0.01 | 0.00 ± 0.03 |
| | MinMax | 0.013 ± 0.001 | 0.03 ± 0.02 | 0.02 ± 0.01 | 0.00 ± 0.03 | 0.00 ± 0.02 |

Table 6: Comparison of Methods in the Non-IID environment with Logistic Regression for 3 seperate runs.Similar to the results in Table 3 under **Non-IID** conditions we have improved tradeoff points using `KFFL`

| | Method | Adult | COMPAS | BANK | ACS | GERMAN |
|---|---|---|---|---|---|---|
| Acc. | FedAvg | 84.35 ± 1.45 | 63.20 ± 1.30 | 94.25 ± 1.15 | 84.10 ± 1.25 | 72.55 ± 1.20 |
| | KFFL | 83.10 ± 0.45 | 62.35 ± 1.35 | 92.20 ± 1.30 | 81.15 ± 1.40 | 75.0 ± 0.01 |
| | KFFL-TD | 83.95 ± 1.10 | 59.55 ± 1.50 | 90.75 ± 1.25 | 81.15 ± 1.10 | 75.0 ± 0.01 |
| | FairFed/FairBatch | 76.30 ± 0.15 | 55.50 ± 1.55 | 88.10 ± 1.35 | 58.75 ± 1.25 | 37.50 ± 1.30 |
| | FairFed-Kernel | 63.45 ± 0.05 | 44.15 ± 1.60 | 88.80 ± 1.20 | 58.80 ± 1.10 | 70.05 ± 1.40 |
| | MinMax | 76.30 ± 0.15 | 56.70 ± 1.45 | 64.95 ± 1.25 | 58.75 ± 1.30 | 70.05 ± 1.15 |
| SPD | FedAvg | 0.17 ± 0.05 | 0.15 ± 0.03 | 0.22 ± 0.02 | 0.09 ± 0.05 | 0.20 ± 0.04 |
| | KFFL | 0.13 ± 0.03 | 0.07 ± 0.02 | 0.06 ± 0.04 | 0.03 ± 0.03 | 0.01 ± 0.05 |
| | KFFL-TD | 0.17 ± 0.04 | 0.06 ± 0.03 | 0.07 ± 0.02 | 0.01 ± 0.04 | 0.01 ± 0.03 |
| | FairFed/FairBatch | 0.005 ± 0.002 | 0.06 ± 0.04 | 0.01 ± 0.03 | 0.01 ± 0.02 | 0.01 ± 0.04 |
| | FairFed-Kernel | 0.35 ± 0.01 | 0.09 ± 0.03 | 0.01 ± 0.04 | 0.01 ± 0.03 | 0.01 ± 0.02 |
| | MinMax | 0.005 ± 0.002 | 0.04 ± 0.02 | 0.20 ± 0.03 | 0.01 ± 0.04 | 0.01 ± 0.03 |
| EO | FedAvg | 0.23 ± 0.04 | 0.24 ± 0.03 | 0.17 ± 0.02 | 0.05 ± 0.03 | 0.03 ± 0.04 |
| | KFFL | 0.13 ± 0.04 | 0.08 ± 0.02 | 0.05 ± 0.03 | 0.03 ± 0.04 | 0.01 ± 0.03 |
| | KFFL-TD | 0.05 ± 0.11 | 0.09 ± 0.03 | 0.06 ± 0.02 | 0.04 ± 0.03 | 0.01 ± 0.04 |
| | FairFed/FairBatch | 0.014 ± 0.002 | 0.10 ± 0.03 | 0.17 ± 0.04 | 0.01 ± 0.02 | 0.01 ± 0.03 |
| | FairFed-Kernel | 0.13 ± 0.01 | 0.06 ± 0.04 | 0.01 ± 0.03 | 0.01 ± 0.02 | 0.01 ± 0.04 |
| | MinMax | 0.014 ± 0.002 | 0.04 ± 0.03 | 0.03 ± 0.02 | 0.01 ± 0.04 | 0.01 ± 0.03 |

Table 7: Comparison of Methods in the Non-IID environment with Neural Network for 3 seperate runs. Similar to the results in Table 3 under Non-IID conditions we have improved tradeoff using `KFFL`

# E    Additional Experiments for Regression Task

Tables 15, 13, 11, 12, 14, and 9 show how `KFFL` performs under various regression tasks.

# F    Fair Regression

The KHSIC method facilitates the training of predictive models for regression tasks while ensuring fairness across various sensitive groups. Fair regression has been extensively explored in centralized settings, with significant contributions from studies such as Chzhen et al. (2020a); Agarwal et al. (2019a); Chzhen et al. (2020b).

| Fairness weight $\lambda$ | RMSE ↓ | KS Difference ↓ |
|:---:|:---:|:---:|
| 0.0 | $0.322400 \pm 0.015698$ | $0.462100 \pm 0.152637$ |
| 1.0 | $0.326067 \pm 0.018071$ | $0.386633 \pm 0.163709$ |
| 50.0 | $0.311633 \pm 0.024243$ | $0.410500 \pm 0.106058$ |
| 100.0 | $0.322400 \pm 0.016542$ | $0.277200 \pm 0.114275$ |

Table 8: RMSE and KS Difference with standard deviations for 5 runs KFFL-TD with **IID** for the Communities and Crime Dataset

| Fairness weight $\lambda$ | RMSE ↓ | KS Difference ↓ |
|:---:|:---:|:---:|
| 0.0 | $0.317528 \pm 0.024826$ | $0.671308 \pm 0.202138$ |
| 1.0 | $0.317952 \pm 0.025052$ | $0.620508 \pm 0.251276$ |
| 5.0 | $0.316936 \pm 0.025075$ | $0.662032 \pm 0.159920$ |
| 50.0 | $0.317576 \pm 0.024314$ | $0.444808 \pm 0.163534$ |
| 100.0 | $0.317592 \pm 0.025111$ | $0.395280 \pm 0.141982$ |

Table 9: RMSE and KS Difference with standard deviations for 5 runs KFFL with **Non-IID** for the Communities and Crime Dataset

In our framework, we employ the Root Mean Squared Error (RMSE) as the primary evaluation metric to assess the accuracy of the regression model. The objective function is defined as:

$$\ell(\mathbf{y}, f(\mathbf{x}; \boldsymbol{\omega})) = \frac{1}{n} \sum_{i=1}^{n} \left( f(\mathbf{x}_i; \boldsymbol{\omega}) - \mathbf{y}_i \right)^2 + \lambda \psi(\boldsymbol{\omega})$$

where $\mathbf{x}_i \in \mathbb{R}^d$ denotes the input features, $\mathbf{y}_i \in \mathbb{R}$ is the target variable, and $f(\mathbf{x}_i; \boldsymbol{\omega})$ represents the model's prediction parameterized by $\boldsymbol{\omega}$. Consistent with the rest of this paper, we exclude the sensitive attribute $\mathbf{s}_i$ from the training process of the fair regressor.

Consider a regression task on a dataset $\mathcal{D} = \{(\mathbf{x}_i, \mathbf{s}_i, \mathbf{y}_i)\}_{i=1}^{n}$, where $\mathbf{x}_i$ is the input feature vector, $\mathbf{s}_i \in \mathcal{S}$ is the sensitive attribute (e.g., gender, race), and $\mathbf{y}_i$ is the target variable (e.g., GPA, income). For each sensitive group $s \in \mathcal{S}$, we define the corresponding subset of data as:

$$\mathcal{D}^s = \{(\mathbf{x}, \mathbf{s}, \mathbf{y}) \in \mathcal{T} : \mathbf{s} = s\}$$

To evaluate fairness in regression tasks, we utilize the Kolmogorov-Smirnov (KS) distance Chzhen et al. (2020a), which measures the distributional differences between the model's predictions for different sensitive groups $s \in \mathcal{S}$. The KS distance is a widely adopted fairness metric in regression, enabling the assessment of disparities between groups. For instance, in a normalized GPA prediction task where the sensitive attribute $\mathcal{S}$ represents gender (e.g., male and female), the KS distance quantifies the difference in GPA predictions between these groups.

The KS distance between predictions for any two groups $s$ and $s'$ is defined as:

$$\mathrm{KS}(f(\mathbf{x}, \mathbf{s})) = \max_{s, s' \in \mathcal{S}} \sup_{t \in \mathbb{R}} \left| F^s(t) - F^{s'}(t) \right|$$

where $F^s(t)$ denotes the empirical cumulative distribution function (CDF) of the model's predictions for group $s$, calculated as:

$$F^s(t) = \frac{1}{|\mathcal{D}^s|} \sum_{(\mathbf{x}, \mathbf{s}, \mathbf{y}) \in \mathcal{D}^s} \mathbb{1}\{f(\mathbf{x}, \mathbf{s}) \leq t\}$$

| Fairness weight $\lambda$ | RMSE $\downarrow$ | KS Difference $\downarrow$ |
|---|---|---|
| 0.0 | $0.320332 \pm 0.024803$ | $0.604128 \pm 0.166035$ |
| 5.0 | $0.320936 \pm 0.024753$ | $0.618280 \pm 0.186275$ |
| 10.0 | $0.321824 \pm 0.024351$ | $0.551812 \pm 0.179296$ |
| 50.0 | $0.314788 \pm 0.026521$ | $0.391712 \pm 0.157367$ |
| 100.0 | $0.318976 \pm 0.023506$ | $0.290872 \pm 0.099638$ |
| 250.0 | $1.034776 \pm 0.015979$ | $0.774960 \pm 0.043567$ |

Table 10: RMSE and KS Difference with standard deviations for 5 runs with KFFL with **IID** for the Communities and Crime Dataset.Optimal points are those with lower RMSE (for accuracy) and KS (for fairness)

| Fairness weight $\lambda$ | RMSE $\downarrow$ | KS Difference $\downarrow$ |
|---|---|---|
| 0.00 | $0.800667 \pm 0.002491$ | $0.450867 \pm 0.065047$ |
| 0.01 | $0.800367 \pm 0.002579$ | $0.398433 \pm 0.224212$ |
| 0.10 | $0.800300 \pm 0.002524$ | $0.412733 \pm 0.303714$ |
| 1.00 | $0.799533 \pm 0.001940$ | $0.148967 \pm 0.009340$ |
| 5.00 | $0.820100 \pm 0.023256$ | $0.348280 \pm 0.030390$ |

Table 11: RMSE and KS Difference with standard deviations for 5 runs KFFL-TD with **IID** for the Law School Dataset

This formulation ensures that the regression model maintains fairness by minimizing the KS distance across all sensitive groups, thereby promoting equitable outcomes.

| Fairness weight $\lambda$ | RMSE ↓ | KS Difference ↓ |
|---|---|---|
| 0.00 | 0.799293 ± 0.002132 | 0.514920 ± 0.124809 |
| 0.01 | 0.799253 ± 0.002306 | 0.452160 ± 0.163044 |
| 0.10 | 0.799187 ± 0.002101 | 0.467133 ± 0.143954 |
| 1.00 | 0.798800 ± 0.002080 | 0.140173 ± 0.032955 |
| 50.00 | 186.105140 ± 120.592723 | 0.195000 ± 0.066063 |

Table 12: RMSE and KS Difference with standard deviations for 5 runs KFFL with **IID** for the Law School Dataset. Optimal points are those with lower RMSE (for accuracy) and KS (for fairness)

| Fairness weight $\lambda$ | RMSE ↓ | KS Difference ↓ |
|---|---|---|
| 0.00 | 0.485100 ± 0.005092 | 0.410600 ± 0.031681 |
| 0.01 | 0.485100 ± 0.005260 | 0.410833 ± 0.026668 |
| 0.10 | 0.484967 ± 0.005424 | 0.408967 ± 0.013403 |
| 1.00 | 0.484767 ± 0.004964 | 0.381000 ± 0.009752 |
| 5.00 | 0.483967 ± 0.004944 | 0.282967 ± 0.010901 |

Table 13: RMSE and KS Difference with standard deviations for 5 runs KFFL-TD with **IID** for the Adult Dataset

| Fairness weight $\lambda$ | RMSE ↓ | KS Difference ↓ |
|---|---|---|
| 0.00 | 0.491523 ± 0.006695 | 0.345782 ± 0.082513 |
| 0.01 | 0.493383 ± 0.008737 | 0.350317 ± 0.111856 |
| 0.10 | 0.493500 ± 0.008616 | 0.361175 ± 0.080166 |
| 1.00 | 0.493492 ± 0.008715 | 0.371725 ± 0.040675 |
| 5.00 | 0.493467 ± 0.009102 | 0.348567 ± 0.034269 |
| 50.00 | 0.494740 ± 0.009975 | 0.130180 ± 0.050417 |
| 100.00 | 0.498000 ± 0.011371 | 0.118580 ± 0.100111 |
| 200.00 | 0.589080 ± 0.194744 | 0.286860 ± 0.115417 |
| 500.00 | 4.510640 ± 5.096420 | 0.351200 ± 0.110467 |

Table 14: RMSE and KS Difference with standard deviations for 5 runs KFFL with **Non-IID** for the Adult Dataset

| Fairness weight $\lambda$ | RMSE ↓ | KS Difference ↓ |
|---|---|---|
| 0.00 | 0.49088 ± 0.009294 | 0.38293 ± 0.016433 |
| 0.01 | 0.49101 ± 0.009303 | 0.39744 ± 0.027059 |
| 0.10 | 0.49053 ± 0.009130 | 0.37377 ± 0.030239 |
| 1.00 | 0.49071 ± 0.009039 | 0.38429 ± 0.053008 |
| 5.00 | 0.49068 ± 0.009474 | 0.26067 ± 0.027564 |
| 50.00 | 0.52160 ± 0.013804 | 0.06012 ± 0.018358 |
| 100.00 | 0.83521 ± 0.015799 | 0.34421 ± 0.018229 |
| 120.00 | 1.10238 ± 0.014253 | 0.43006 ± 0.019893 |
| 500.00 | 40.32616 ± 48.938383 | 0.27088 ± 0.139738 |

Table 15: RMSE and KS Difference with standard deviations for 5 runs KFFL with **IID** for the Adult Dataset

# G   Communication-Efficient Kernel Regularized Fair Learning

In this section, we address the computational and communication challenges of incorporating kernel-based fairness regularizers into federated learning. Specifically, we leverage Random Feature Maps (RFMs) to approximate kernel functions efficiently, reducing both computational complexity and communication overhead.

## G.1   Random Feature Maps for Kernel Approximation

Kernel methods are powerful tools in machine learning but often suffer from high computational complexity, especially when dealing with large datasets. Computing kernel matrices requires $\mathcal{O}(n^2)$ memory and $\mathcal{O}(n^3)$ computational time, which is impractical for large $n$.

To overcome this, Rahimi & Recht (2007) introduced Random Feature Maps (RFMs) to approximate shift-invariant kernel functions. A shift-invariant kernel $\kappa(\mathbf{x}, \mathbf{y}) = \kappa(\mathbf{x} - \mathbf{y})$ can be represented using the Fourier transform via Bochner's theorem. Specifically, the kernel can be expressed as:

$$\kappa(\mathbf{x}, \mathbf{y}) = \int_{\mathbb{R}^d} p(\boldsymbol{\omega}) e^{j\boldsymbol{\omega}^\top (\mathbf{x}-\mathbf{y})} d\boldsymbol{\omega}, \tag{29}$$

where $p(\boldsymbol{\omega})$ is the spectral density function of the kernel $\kappa$.

**Constructing Random Feature Maps**   To approximate $\kappa(\mathbf{x}, \mathbf{y})$, we draw $D$ random samples $\{\boldsymbol{\omega}_k\}_{k=1}^D$ from $p(\boldsymbol{\omega})$ and define the random feature map $\phi : \mathbb{R}^d \to \mathbb{R}^D$ as:

$$\phi(\mathbf{x}) = \sqrt{\frac{2}{D}} \left[ \cos(\boldsymbol{\omega}_1^\top \mathbf{x} + b_1), \ldots, \cos(\boldsymbol{\omega}_D^\top \mathbf{x} + b_D) \right], \tag{30}$$

where $\{b_k\}_{k=1}^D$ are drawn uniformly from $[0, 2\pi]$.

With this mapping, the kernel function can be approximated as:

$$\kappa(\mathbf{x}, \mathbf{y}) \approx \phi(\mathbf{x})^\top \phi(\mathbf{y}). \tag{31}$$

**Frequency of Drawing Random Features**   In our implementation, the random features are drawn **once at the beginning** of global training round and are fixed throughout the optimization process. This ensures consistency across iterations and clients, and avoids the overhead of regenerating random features at each iteration.

## G.2   Computing $\mathbf{Z}_s$ and $\mathbf{Z}_f$

In the context of our fairness regularizer, we need to compute feature maps for both the sensitive attributes $S$ and the model outputs $f_{\boldsymbol{\omega}}(\mathbf{X})$. Specifically:

- **Sensitive Attributes Feature Map ($\mathbf{Z}_s$):** For each data point $i$, we compute $\phi(s_i)$, where $s_i$ is the sensitive attribute of the $i$-th sample. The matrix $\mathbf{Z}_s \in \mathbb{R}^{n \times D}$ has rows $\phi(s_i)^\top$.

- **Model Outputs Feature Map ($\mathbf{Z}_f$):** For each data point $i$, we compute $\phi(f_{\boldsymbol{\omega}}(\mathbf{x}_i))$, where $f_{\boldsymbol{\omega}}(\mathbf{x}_i)$ is the model output (e.g., logits) for the $i$-th sample. The matrix $\mathbf{Z}_f \in \mathbb{R}^{n \times D}$ has rows $\phi(f_{\boldsymbol{\omega}}(\mathbf{x}_i))^\top$.

**Example: Gaussian Kernel Approximation**   As an example, consider the Gaussian (RBF) kernel:

$$\kappa(\mathbf{x}, \mathbf{y}) = \exp\left( -\frac{\|\mathbf{x} - \mathbf{y}\|_2^2}{2\sigma^2} \right). \tag{32}$$

The spectral density of the Gaussian kernel is $p(\boldsymbol{\omega}) = \mathcal{N}(\boldsymbol{\omega}; \mathbf{0}, \sigma^{-2}\mathbf{I})$. Therefore, to approximate the Gaussian kernel, we draw $\boldsymbol{\omega}_k \sim \mathcal{N}(\mathbf{0}, \sigma^{-2}\mathbf{I})$ and compute the feature maps as:

$$\phi(\mathbf{x}) = \sqrt{\frac{2}{D}} \left[ \cos(\boldsymbol{\omega}_1^\top \mathbf{x} + b_1), \ldots, \cos(\boldsymbol{\omega}_D^\top \mathbf{x} + b_D) \right]. \tag{33}$$

**Orthogonal Random Features**  To improve the quality of the approximation and reduce variance, we employ Orthogonal Random Features (ORF) as proposed by Yu et al. (2016). Instead of sampling $\boldsymbol{\omega}_k$ independently, we construct them to be orthogonal, which can lead to better kernel approximations with fewer features.

