# OpenReview forum: "Group Fair Federated Learning via Stochastic Kernel Regularization"
_TMLR — Accepted by TMLR_

### Review · Reviewer_JkPu · 2025-02-05

**Summary Of Contributions:**

This paper studies group fairness in federated learning by introducing the Kernel Hilbert-Schmidt Independence Criterion (KHSIC) as a fairness regularizer in the FL framework. To further reduce computational and communication overheads, Kernel Fair Federated Learning (KFFL) is achieved using random feature map approximation. The authors provide theoretical analysis, and experimental results on both IID and non-IID settings demonstrates the effectiveness of the proposed method.

**Audience:**

Yes

**Claims And Evidence:**

Yes

**Requested Changes:**

I suggest that the authors address the weaknesses if necessary.

**Strengths And Weaknesses:**

**Strengths**:
1. The paper is overall well-written and well-structured. The basic concepts related to group fairness in FL have been appropriately introduced and are easy to follow.
2.Experimental results seem solid and convincing, and various settings and datasets are considered to provide experimental validation.

**Weaknesses**:
1. I think the overall motivation for applying KHSIC for group fairness is not very convincing. This is because we know that KHSIC may lead to extra communication and computation costs (which the authors address by using random feature maps). Therefore, I find the motivation not particularly compelling.
2. Figure 2 seems somewhat unclear. Based on Algorithm 2, it appears that the FAIR1, FAIR2, and DATA rounds do not occur in the same round. However, in Figure 2, it looks as if they happen simultaneously. Moreover, why do the authors choose DATA as a component in both the algorithm and the figure? I think this may lead to some misunderstanding, namely that clients need to send data to the server in certain rounds, which could violate a key constraint in FL.
3. Table 2 seems confusing. Why are the rounds of communication different for each method?
4. Can the authors also provide an exact computation time comparison for the baselines?
5. Regarding Table 6, based on which rule is bold formatting applied? For example, for the SPD metric, it seems that KFFL and KFFL-TD are neither the largest nor the smallest SPD scores.

---

> ### Author Response · Authors · 2025-02-28
>
> ``` Why use KHSIC if it adds extra communication and computational overhead? Is it really worth it?```
>
> This choice depends on the practitioner's considerations.  KFFL introduces a local step to compute $Z_s^\top Z_f$, an $O(D^3 + DB)$ operation, where $B$ is the minibatch size. With $D \ll n$ (e.g., $D = 10, B = 64$ in our experiments), this is negligible compared to backpropagation, thus the method adds little computational overhead. It does, however, introduce additional communication in exchange for a principled statistical dependence measure.
>
> KHSIC quantifies dependencies between random variables. A low KHSIC between $f(x; \omega)$ and $s$ implies approximate statistical parity (see Theorem 2 of [1]), so using it as a regularizer allows a theoretically grounded trade-off between fairness and accuracy. We will emphasize this theoretical advantage by citing and discussing Theorem 2 from [1] in more detail than currently is provided.
>
> Our experiments confirm this advantage, showing KFFL can better explore trade-offs between fairness and accuracy as $\lambda$ varies, compared to baselines varying their fairness parameters.
>
> Thus, KFFL is a strong option for practitioners needing full trade-off flexibility.
>
> ```Figure 2 suggests these sub-rounds happen simultaneously. Also, ‘DATA’ sounds like raw data is uploaded, which violates FL. ```
>
> The figure depicts three communication phases (FAIR1, FAIR2, DATA) per global iteration, executed sequentially, not simultaneously. We will clarify the ordering in the figure.
>
> The only communicated quantities are:
> - Model parameters
> - Small $D \times D$ kernel-statistic matrices
> - Gradients
>
> Sensitive features remain local. “DATA” refers to each client’s local update phase; no raw $\{x_i, s_i\}$ is sent to the server.
>
> To avoid confusion, we will rename "DATA" to "Local Update" and explicitly note that no raw data leaves the client.
>
> ```Table 2 is unclear about why communication rounds differ for each method. ```
>
> Some methods require multiple sub-rounds per global iteration. FedAvg has one, while fairness-aware methods (e.g., KFFL, FairFed) need extra sub-rounds for fairness statistics or gradients. Table 2 compares KFFL’s additional communication with baselines.
>
> KFFL has three sub-rounds (FAIR1, FAIR2, Local Update). KFFL-TD reduces this to two. FairFed, a key baseline, also has three.
>
> ```Where is the runtime comparison to baselines? ```
>
> Our goal is to achieve principled and flexible fairness-accuracy trade-offs while reducing the communication overhead (from $n \times n$ to $D \times D$ matrices, where $D=10$ in our experiments). Computing random feature maps takes $O(DB)$ time (where $B$ is batch size), and matrix multiplication for the fairness term takes $O(D^3)$. These computation costs are negligible compared to local backpropagation, and all methods use SGD type local updates, so we did not provide wall clock comparisons.
>
> Federated methods are communication-bound so are usually compared by communication cost (# of messages/bits) rather than local computation costs. As noted in Appendix B.1, all methods converged after 10 global epochs.
>
> ``` Why do you bold certain KFFL results if they’re not strictly the min or max? ```
>
> We bolded results to highlight desirable balances of accuracy and fairness, as a method may not minimize one metric but still achieve the best trade-off overall. To be more objective, we will remove the bolding and update the table caption to explain the insights from the experimental results directly.
>
> [1] Kevin Kim and Alex Gittens. Learning fair canonical polyadical decompositions using a kernel independence criterion. arXiv preprint arXiv:2104.13504, 2021.

---

### Review · Reviewer_c3nu · 2025-02-12

**Summary Of Contributions:**

The paper introduces Kernel Fair Federated Learning (KFFL), a framework aimed at ensuring group fairness in federated learning (FL).

The key innovation is the use of the Kernel Hilbert-Schmidt Independence Criterion (KHSIC) as a fairness regularizer.

 Since computing KHSIC directly is computationally expensive and communication-heavy in FL, the authors propose using Random Feature Maps (RFMs) to approximate KHSIC efficiently. T

o handle the resulting non-convex optimization problem, they introduce FedProxGrad, a federated proximal gradient algorithm that provides convergence guarantees.

**Audience:**

Yes

**Claims And Evidence:**

Yes

**Requested Changes:**

**Critical (Necessary for Acceptance)**
1. Clarify Algorithmic Novelty
    * Explain what is new about FedProxGrad or conform that it is simply applying FedProx to a regularized problem.
2. Correct Definition 2 and Convergence Analysis
    * Properly cite the correct (G, B)-Bounded Dissimilarity definition.
    * Provide a convergence rate analysis for different values of $B$, particularly when $B = \infty$.
3. Include a Hyperparameter Sensitivity Analysis
    * Conduct experiments analyzing the impact of $\lambda$ on fairness-accuracy tradeoffs.
4. Add Related Work Discussion on Random Feature Methods
    * Discuss the similarities and differences between KFFL and A Random Projection Approach to Personalized Federated Learning.
5. Clarify and Improve the Explanation of the Pareto Frontier
    * Clearly define what the Pareto frontier represents in the context of FL fairness.
    * Improve the readability of Figure 3, clarify the meaning of the solid curves, and ensure that the figure is analyzed in the appropriate section.

**Recommended (Would Strengthen the Paper)**

6. Include a Runtime Comparison to Baselines
    * Report training times for KFFL and FedProx across different FL settings.
7. Explain Why the Model Uses Only Non-Sensitive Covariates ($x$) as Inputs
    * It is unclear why $s$ is excluded—a common belief is that more features improve performance.
8. Clarify Why Traditional Fairness Metrics Are Not Directly Applicable in FL
    * The paper states that Equalized Odds (EOD) and Statistical Parity Difference (SPD) are not directly applicable in FL (Page 4).
    * Why? What challenges does FL introduce that make these fairness measures difficult to use?

**Strengths And Weaknesses:**

**Strengths**
1. Addresses an Important Challenge in FL
    * The paper tackles the well-known problem of ensuring fairness in federated learning while maintaining efficiency.
2. Theoretical Soundness and Convergence Analysis
    * Using KHSIC as a fairness measure is a strong theoretical choice, capturing non-linear dependencies effectively.
    * The mathematical formulation is solid, and the convergence analysis is well presented.
3. Communication Efficiency through Random Feature Maps
    * Leveraging RFMs to approximate KHSIC significantly reduces communication costs, making it feasible for FL.
    * The communication efficiency analysis is well explained and highlights the advantages of KFFL over naive kernel methods.
4. Comprehensive Empirical Evaluation
    * The experiments compare KFFL against strong baselines (FedAvg, FairFed, MinMax) on classification and regression tasks.
5. Generally Well-Written Paper
    * The paper is mostly well-structured and readable, though some citations are missing, and certain word choices create confusion (see Weaknesses).


**Weaknesses**
1. Limited Novelty in Algorithm Design
    * The algorithm is not fundamentally new—it is essentially FedProx applied to a regularized problem.
    * While the combination of KHSIC regularization and RFMs is interesting, it does not introduce a fundamentally novel optimization framework.
2. Misrepresentation of Definition 2 and Convergence Rate Issues
    * The (G, B)-Bounded Dissimilarity condition is not exactly the one used by Li et al. (2020) and Karimireddy et al. (2020). The authors should cite the correct work and clarify the assumptions.
    * Theorem 1 states a $1/T$ convergence rate, but the standard worst-case rate for FedProx is $1/\sqrt{T}$ (see On Convergence of FedProx: Local Dissimilarity Invariant Bounds, Non-smoothness, and Beyond, arXiv:2206.05187). Could the author explain the difference?
    * The assumption that $B$ is finite is not always valid. The authors should provide a more systematic convergence rate analysis for different values of $B$, especially when $B=\infty$.
3. Lack of Discussion on Related Work in Random Feature Approximations
    * The idea of using random features to reduce communication costs is not new.
    * A relevant paper, A Random Projection Approach to Personalized Federated Learning: Enhancing Communication Efficiency, Robustness, and Fairness (JMLR 2023), also uses a random matrix and studies fairness but is not cited in this paper. A discussion with it is preferred.
4. Computational Overhead Not Analyzed
    * While KFFL reduces communication costs, it still adds computational overhead due to KHSIC and its gradient computation.
    * The paper lacks a runtime analysis comparing the wall-clock time of KFFL to baseline methods.
5. Hyperparameter Sensitivity Not Addressed
    * The choice of the fairness weight $\lambda$ is crucial, but its impact is not well-analyzed.
    * How should practitioners select $\lambda$ in practice? A sensitivity analysis would be valuable.
6. Unclear Definition and Analysis of "Pareto Frontier"
    * The term “Pareto frontier” is mentioned multiple times in Section 6.1, but its exact definition is unclear.
    * Is it simply a curve in the plot of group fairness vs. accuracy?
    * Figure 3 is somewhat hard to read, and its analysis is far away from the figure.
    * The meaning of the solid curves is unclear—are these the so-called Pareto frontiers?

---

> ### Author Response · Authors · 2025-02-28
>
> ```The algorithm is basically FedProx on a composite regularized problem; combining KHSIC and RFMs might not be a fundamental algorithmic innovation.```
>
> FedProxGrad differs significantly from FedProx. FedProx is a federated proximal point algorithm designed to minimize a single objective $F = \sum_{i=1}^k F_i$ using a federated proximal point algorithm. FedProxGrad is a federated proximal gradient algorithm designed to minimize a composite objective, \( F + G \), where _only_ $F$ can be written as a sum of client-level objectives, and $G$ may be arbitrary.
>
> We designed FedProxGrad because our KHSIC fairness regularizer results in $F + G$, where $F$ is client-decomposable but $G$ is not. Thus, __FedProx cannot be used for our formulation of fairness-regularized federated learning__.
>
> Existing federated composite optimization (FCO) methods, such as those cited in Section 3, impose convexity assumptions on $F$ and/or $G$, which FedProxGrad does not require. Thus FedProxGrad is a novel, more generally applicable, FCO algorithm with convergence guarantees. We note that it was not until recently that the convergence of stochastic proximal gradient algorithms for fully nonconvex composite optimization problems were shown to be convergent in the shared memory setting [3]. We have accomplished the same in the federated setting.
>
> While the constituent parts of our algorithms (RFMs, KHSIC) each exist in prior literature, their combined formulation to obtain a principled approach to fair federated learning— backed by the a novel FCO algorithm that overcomes the aforementioned limitations of previous FCO methods- goes beyond a simple extension of any prior work.
>
> We thank the reviewr for the clarifying question and will add the above discussion to the paper.
>
> ```It differs from Li et al. (2020) and Karimireddy et al. (2020). Please cite the correct work and clarify the assumptions.```
>
> We use the $(G, B)$-bounded gradient dissimilarity definition given in Karimireddy et al. (2020) as (A1). Namely, when $ f = \frac{1}{N} \sum_{I=1}^N f_i $,
>
> $\frac{1}{N} \sum_{i=1}^N \|\nabla f_i(\omega)\|_2^2  \leq G^2 + B^2 \|\nabla f(\omega)\|_2^2.$
>
> For our composite objective $ F = \ell(\omega) +  \phi(\omega)  = \frac{1}{N} \sum_{I=1}^N [\ell_i(\omega) + \phi(\omega) ]  $, this takes the form stated in our work:
>
> $\frac{1}{N} \sum_{i=1}^N \|\nabla \ell_i(\omega) + \nabla \phi(\omega) \|_2^2  \leq G^2 + B^2 \|\nabla F(\omega)\|_2^2.$
>
> We will clarify this application in the paper, and specifically cite the definition used as (A1) in Karimireddy et al. (2020).
>
> ```There is a known result that FedProx can achieve only $1/\sqrt{T}$ under certain conditions. Could the authors explain the discrepancy?```
>
> Li et al., 2020 guarantees a convergence rate of $1/T$ when the objective is a general nonconvex function and the objective function is $(G,B)$-BGD with $G = 0$. This assumption implies in particular that the globally optimal model is optimal for each client.
>
> Our analysis permits $G > 0$ and achieves a rate of $1/T$. This assumption is more realistic, as clients' optimal models can differ from the globally optimal model. Yuan and Li take the even more general approach of making no BGD assumption. Under the conditions of Lipschitzianity of the objective function and additive suboptimality of the local iterates, they find a convergence rate of $1/\sqrt{T}$.
>
> Thank you for the pointer to (Yuan and Li, 2022): we will cite it as a potential avenue for removing the BGD assumption altogether in future work.
>
> ```What if $B$ is not finite? The authors should analyze that scenario more systematically.```
>
> Following Karimireddy et al. (2020), we assume bounded dissimilarity so local progress leads to global progress. If $B \rightarrow \infty$, Theorem 1 implies the global step size should decrease like $O(1/B^2)$, slowing convergence.
>
> This mirrors common optimization assumptions (e.g., bounded Lipschitz or bounded Hessian assumptions): when violated, step sizes shrink, and rates degrade. We thank the reviewer for raising this point, and will expand our discussion of Theorem 1 to include these considerations.
>
> ```Methods using random projections to reduce communication are not new. Please discuss and cite relevant work.```
>
> We agree! Our contribution is in applying random projections to achieve fair federated learning in a statistically principled manner by communicating a $D \times D$  matrix instead of an $ n \times n$ matrix (in our experiments, $D=10$).
>
> We will cite and compare to the JMLR 2023 reference and similar random-projection FL approaches (e.g. [1], [2]). These works employ random projections for other purposes than achieving group fairness.  In particular, the JMLR work uses randomized projections to help in their aim of achieving _performance fairness_ (see Definition 1 of the paper), by minimizing the variance in the client objectives; this concept of fairness differs from that of group fairness, which is our focus.

---

> ### Author Response · Authors · 2025-02-28
>
> ```Even if KFFL reduces communication, the kernel-based fairness computation adds local overhead. Is there a runtime or wall-clock comparison?```
>
> KFFL introduces a local step to compute $Z_s^\top Z_f$, an $O(D^3 + DB)$ operation, where $B$ is the minibatch size. With $D \ll n$ (e.g., $D = 10, B = 64$ in our experiments), this is negligible compared to backpropagation.
>
> Since FL performance is typically communication-bound, and all methods require similar local operations, we did not conduct a runtime comparison.
>
> ```$\lambda$ is central to controlling fairness vs. accuracy. How do we pick it in practice? A sensitivity or ablation is needed.```
>
> Practitioners must tune fairness regularization parameters (e.g., $\lambda$) of any method to balance fairness and accuracy to their needs. Our method smoothly traces this trade-off (Figures 3, 4, Table 3), making the $\lambda$ selection systematic.
>
> We recommend binary search on $\lambda$ to find an acceptable performance point.
>
> ```The paper repeatedly uses ‘Pareto frontier’ in Section 6.1. Is it just the accuracy–fairness trade-off curve? Figure 3 is difficult to parse.```
>
> Yes, we mean the accuracy-fairness trade-off curve. We will change the text and figures in the paper to use the more accurate term "accuracy-fairness trade-off curve."
>
> [1] Zong et al. "Privacy by projection: Federated population density estimation by projecting on random features." PoPETs, 2023.
> [2] Asi et al. "Fast optimal locally private mean estimation via random projections." NeurIPS, 2023.
>
> [3] Xu, Y. et al. Non-asymptotic analysis of stochastic methods for non-smooth non-convex regularized problems. NeurIPS, 2019.

---

### Review · Reviewer_KtdY · 2025-02-16

**Summary Of Contributions:**

This paper proposes Kernel Fair Federated Learning (KFFL), a framework for ensuring group fairness in federated learning (FL). The authors use the Kernel Hilbert-Schmidt Independence Criterion (KHSIC) as a fairness regularizer to minimize statistical dependence between model predictions and sensitive attributes (e.g., race, gender). To address the high computational and communication costs of kernel-based methods, the authors approximate KHSIC using Random Feature Maps (RFM). They also introduce FedProxGrad, a federated proximal gradient algorithm, to handle the resulting non-convex optimization problem while ensuring convergence.

**Audience:**

Yes

**Claims And Evidence:**

No

**Requested Changes:**

See above

**Strengths And Weaknesses:**

Strengths:
- This paper proposes Kernel Fair Federated Learning (KFFL), incorporating the Kernel Hilbert-Schmidt Independence Criterion (KHSIC) as a fairness regularizer. This is one of the first applications of KHSIC to federated learning (FL), providing a novel approach to enforcing group fairness in FL.
- The authors propose using Random Feature Maps (RFM) to address the high computational and communication costs associated with kernel-based methods.
- The paper presents comprehensive experiments on both IID and Non-IID federated settings, using multiple datasets.
- The authors also present the Pareto frontier, which helps in understanding the trade-offs between fairness and performance.

Weakness:
- To derive equation 8, the sensitive information $s$ needs to be clearly defined. However, In image-based tasks or unstructured data, sensitive attributes are not explicitly labeled. For instance, the skin tone and gender of a person in the image could be the sensitive information in image-based tasks. This makes KFFL hard to apply in real-world cases where fairness-sensitive features are implicit.
- As the authors mentioned, $s_i$ contains sensitive covariates (which may be a binary scalar $s_i$ or multi-dimensional vector $s_i$). However, if I understand correctly, $s_i$ is always chosen as a binary vector in the experiments, even if it could actually be both race and sex in the ADULT dataset. Could the authors explain the reasoning behind this choice?
- In Sections 6.2 and 6.3, the authors select specific trade-off points to showcase the performance of KFFL. However, if I understand correctly, FairFed and other methods do not explore the Pareto frontier. This may lead to an unfair comparison, where KFFL appears superior simply because other methods were not evaluated across the full Pareto frontier.
- Typically, adding fairness constraints reduces accuracy, but Tables 4 & 5 show no clear drop in accuracy when increasing fairness regularization, especially considering the standard deviation. This may indicate that the dataset does not contain strong predictive features tied to the sensitive attribute. I would appreciate it if the authors could explain why accuracy remains almost stable, as this is important for understanding the real impact of fairness constraints.
- In practical applications, companies often have a predefined fairness constraint (e.g., "We need SPD ≤ 0.05"), rather than needing to explore the entire Pareto frontier. Therefore, while Pareto frontier analysis is useful for research, it may not be necessary for practical applications. Could the authors elaborate on this? Additionally, the paper could discuss how to efficiently select a fairness constraint without requiring full Pareto exploration.
- As the authors themselves mentioned, they only consider full client participation. I am wondering if this choice is purely due to computational cost and/or time constraints.
- In Figures 3 and 4, how are the curves for KFFL and KFFL-TD drawn? It appears to be an interpolation between points—please correct me if I’m wrong. Additionally, it seems that FairFed outperforms KFFL-TD in some parts of the Low SPD Region and Low EOD Region. If the fairness budget were chosen differently, would FairFed consistently outperform KFFL-TD?

Minor:
- In Figure 1 (bottom right), the scale still shows that one weighs more than another even though the text shows Does not discriminate against gender.

---

> ### Author Response · Authors · 2025-02-28
>
> ```KFFL requires explicit knowledge of the sensitive attribute. In tasks like images or unstructured data, they may not be labeled or may be implicit. Isn’t this a limitation in real-world applications?```
>
> Sensitive attributes are required at training time to enforce fairness, just as ground truth labels are needed for accuracy. They are not used at test time.
>
> This assumption is standard in both centralized and federated fairness approaches [1-4], and benchmark datasets provide these attributes. Some work addresses fairness without sensitive attributes at training time, but that is beyond our scope.
>
> ```Why do you only use one protected attribute (e.g., race or sex) when some datasets, like ADULT, could use both?```
>
> We follow the standard fairness benchmark protocol, which uses a single binary attribute per experiment, ensuring straightforward comparisons with existing baselines (e.g., FairFed [1], MinMax [2]) that also focus on a single sensitive feature.
>
> However, our KHSIC-based approach is not restricted to a single attribute—multiple protected features can be incorporated without modifying KFFL’s algorithm.
>
> ```In Section 6.2 and 6.3, you vary the parameter in KFFL to map out a full accuracy–fairness Pareto frontier, but baselines like FairFed or MinMax do not naturally produce such smooth frontiers. Is that a fair comparison?```
>
> For all methods, we varied hyperparameters that control fairness-accuracy trade-offs (e.g., FairFed’s “fairness budget”) to obtain multiple (accuracy, fairness) points. We report all resulting points (including from the baselines) for completeness. The fairness hyperparameters for the baseline methods, and the choice of ranges they are swept over, is discussed in Appendix B.1.
>
> The results show FairFed and MinMax produce more clustered points, whereas KFFL forms a smooth trade-off curve. This suggests KFFL allows more precise and reliable control over fairness-accuracy trade-offs.
>
> ```Tables 4 & 5 show very little accuracy drop even with higher fairness regularization in regression. Does this imply the datasets are not that sensitive to fairness constraints?```
>
> We focused on a range of $\lambda$ values that produce balanced fairness with minimal accuracy loss in the results shown, as that is the operating regime most of practical interest. Pushing $\lambda$ higher results in more significant accuracy drops. We will clarify this and include results at extreme $\lambda$ values.
>
> ```Real-world applications often fix a single fairness constraint (e.g., SPD), rather than exploring the whole frontier. How does KFFL help in such a scenario?```
>
> Both KFFL and the baselines use hyperparameters to balance fairness and accuracy. However, baseline methods may not achieve all SPD thresholds, whereas KFFL’s smooth frontier makes it more likely to meet a fixed constraint by adjusting $\lambda$.
>
> The important practical implication is that some SPD/accuracy trade-offs may not be achievable for any hyperparameter values for the baseline methods, while they may be for KFFL.
>
> ```You assume all clients participate every round. Is this only for simplicity, or is there a deeper reason?```
>
> This assumption simplifies analysis. Our focus was on the trade-offs between fairness and accuracy. Extending our guarantees to partial participation would add complexity without adding new insights beyond what is known about full versus partial participation in existing federated learning literature. FairFed [1] and MinMax [2] make similar assumptions.
>
> ```How did you generate the KFFL/KFFL-TD curves in Figures 3 and 4? Sometimes FairFed outperforms KFFL in certain regions—could different tuning for FairFed make it consistently better?```
>
> Each KFFL curve represents multiple runs at different $\lambda$ values, interpolated for visualization. FairFed and MinMax generate sparse, uneven points when varying their hyperparameters.
>
> While FairFed could be better at some isolated point (e.g., achieving extremely strict fairness in exchange for a large accuracy drop), it does not consistently dominate KFFL on the entire curve. KFFL's main advantage is that it can reach many points on the trade-off, rather than only a few.
>
> ```In Figure 1 (bottom right), the scale still shows that one weighs more than another even though the text shows ‘Does not discriminate against gender.'```
>
> Thank you for identifying this issue. We will correct this figure in the final paper.
>
> [1] Ezzeldin et al. FairFed: Enabling Group Fairness in Federated Learning. AAAI, 2023.
> [2] Papadaki et al. Minimax Demographic Group Fairness in Federated Learning. FAccT, 2022.
> [3] Baharlouei et al. Rényi Fair Inference. arXiv:1906.12005, 2019.
> [4] Pérez-Suay et al. Fair Kernel Learning. ECML PKDD, 2017.

---

### Author Response · Authors · 2025-02-28
**Thank you to the reviewers**

Thank you for taking the time to review our work and provide thoughtful suggestions for improvement. We have carefully addressed your questions and critiques, and found this process helpful in strengthening the positioning of our work. We appreciate your consideration of our responses as you make your final recommendation. Thank you!

---

### Decision · Action_Editor_QDyM · 2025-04-12

**Recommendation:** Accept with minor revision

**Comment:**

The submission has received expert reviews from the domain who provided detailed and technical comments for the authors to further improve the submission. Most of the comments from Reviewer c3nu and JkPu have been properly addressed by the authors during the rebuttal period, including clarification on the discrepancy of convergence rate with existing work, as well as citation/discussion of related work in using random Fourier features to reduce the communication costs.

However, there are several comments from KtdY, mostly regarding experiments, have not been well addressed by the authors. In particular, despite the claim of KHSIC being able to work in multi-group or multiple sensitive attributes setting, the experiments in the paper only include binary groups. Furthermore, I also concur with the reviewer's comments on the presented Pareto-front: at least in the large $\lambda$ regime, I would anticipate a drop in accuracy to compensate for fairness. The authors need to address the above two points in their revision:

- Add at least one more dataset in the experiment where there are multiple sensitive attributes (gender and race for example) or a single sensitive attribute with more than 2 groups.

- Show a more wholistic view of the Pareto-front by also including large values of $\lambda$.

In terms of novelty, from a technical perspective, there is a consensus from the reviewers that novelty may not be enough to warrant a journal-to-conference track presentation -- the use of KHSIC to enforce statistical independence is not new in centralized setting; the use of random feature maps to avoid the explicit computation of kernel matrices is also not new; That being said, the overall presentation of the paper is clear and sound, hence the overall decision of accept with minor revision but no journal-to-conference presentation.

**Audience:**

Ensuring fairness in federated learning is an important and relevant topic for the broader machine learning community. This paper’s focus on bridging theoretical rigor (via KHSIC-based fairness) and practical constraints (federated setting, communication overhead) should be of interest to researchers and practitioners alike.

**Claims And Evidence:**

The main contribution of the submission is an algorithm that uses kernel Hilbert-Schmidt Independence Criterion (KHSIC) in the federated learning setting to ensure group fairness. Due to the need of communication between clients and the centralized server, the authors proposed a proximal gradient descent algorithm to tackle this problem, by avoiding the explicit computation of the kernel matrix and using random feature maps as an approximation. Theoretically, the authors provide an $1/T$ convergence rate in terms of squared gradient norms, i.e., first-order stationarity condition. Empirically the authors showed that by choosing different regularization hyperparameters, the proposed approach can efficiently explore the Pareto front between accuracy and group fairness.